# Auditory-motor synchronization and perception suggest partially distinct time scales in speech and music

Alice Vivien Barchet [1✉], Molly J. Henry[2,3], Claire Pelofi[4,5] & Johanna M. Rimmele [1,5✉]

Speech and music might involve specific cognitive rhythmic timing mechanisms related to differences in the dominant rhythmic structure. We investigate the influence of different motor effectors on rate-specific processing in both domains. A perception and a synchronization task involving syllable and piano tone sequences and motor effectors typically associated with speech (whispering) and music (finger-tapping) were tested at slow (~2 Hz) and fast rates (~4.5 Hz). Although synchronization performance was generally better at slow rates, the motor effectors exhibited specific rate preferences. Finger-tapping was advantaged compared to whispering at slow but not at faster rates, with synchronization being effector-dependent at slow, but highly correlated at faster rates. Perception of speech and music was better at different rates and predicted by a fast general and a slow finger-tapping synchronization component. Our data suggests partially independent rhythmic timing mechanisms for speech and music, possibly related to a differential recruitment of cortical motor circuitry.

---

[1] Department of Cognitive Neuropsychology, Max Planck Institute for Empirical Aesthetics, Frankfurt am Main, Germany. [2] Research Group 'Neural and Environmental Rhythms', Max Planck Institute for Empirical Aesthetics, Frankfurt am Main, Germany. [3] Department of Psychology, Toronto Metropolitan University, Toronto, Canada. [4] Music and Audio Research Laboratory, New York University, New York, NY, USA. [5] Max Planck NYU Center for Language, Music, and Emotion, New York, NY, USA. ✉email: alice-vivien.barchet@ae.mpg.de; johanna-rimmele@ae.mpg.de

There exists a long-held debate about the shared nature versus the specificity of the mechanisms involved in speech and music processing[1–11]. Speech and music perception and production are uniquely human behaviors and the produced signals share characteristic features, such as their inherently rhythmic and hierarchical structure[12]. At a closer look, however, speech and music signals exhibit a distinct rhythmic profile and differences in rate-specific processing[13–15]. It is unclear whether speech and music recruit distinct cortical motor timing mechanisms related to the motor effectors commonly used in both domains[16,17]. In an auditory perception task and a perception-production synchronization task, we probe rate-specific processing and its modulation by the use of different motor effectors addressing the question of interdependencies of rhythmic timing in speech and music processing.

Speech and music signals display an inherently (quasi-)rhythmic structure, which is one of the characteristics that has been suggested to drive the structural and mechanistic overlap between speech and music processing[7,13]. Humans take advantage of this signal property for making temporal predictions and for event segmentation[18–25]. More specifically, the temporal processing of rhythmic structure in speech and music has been related to endogenous brain rhythms that show rhythmicity in the same frequency range as the speech and music signals[16,26–31]. While it is still debated whether such brain rhythms emerged from the natural properties of speech and music or whether rhythm in speech and music evolved around this functional cortical architecture[32], a functional relevance has been proposed. Endogenous brain rhythms may support predictive processing and event segmentation by entraining to the rhythmic temporal modulations in the speech[20,33–35] and the music signal[18,36–38]. Speech research emphasized the role of auditory cortex brain rhythms in the theta range (~4.5 Hz) that are proposed to constrain temporal processing[20,22,39,40]. Additionally, an impact of rhythmic prediction from the motor system has been discussed[41,42]. The motor system involvement in rhythmic timing is in accordance with a - for obvious biological reasons - tight coupling of sensory and motor systems in the speech and in the music domain. The motor regions involved in production have been shown to be activated solely by listening to speech[43,44] and music[45–47]. Temporal motor prediction has been shown to support speech processing in demanding listening conditions[48,49], as well as music processing[17,31,50]. The supplementary motor area and the basal ganglia have been suggested to function as a pacemaker during speech perception[51,52] and particularly during beat perception and anticipation in music[17,53,54]. Particularly, slow delta brain rhythms around 2 Hz observed in the supplementary motor area seem to be involved in temporal predictions provided by the motor system[26,29,31,55]. This time scale corresponds to the time scale of beats in music[17,31,56–58], while its role in speech processing is not fully understood. However, delta brain rhythms around 1–2 Hz have been suggested to support domain-general rhythmic motor timing[17,31]. In summary, speech and music processing rely on the signals' inherently rhythmic structure and overlapping brain areas including the motor system are involved in their processing.

In spite of their considerable overlap, the produced speech and music signal show crucial differences in rhythmic characteristics. Analyses of large corpora of produced speech and music signals revealed that for diverse types of music played on various instruments, slow acoustic amplitude modulations around 1–2 Hz are dominant[13,15]. Interestingly, this rate corresponds to the preferred rate of human beat perception[59,60], and beat perception has no equivalent in the speech domain[61]. Although the beat might be crucial for interpersonal coordination in musical ensembles[62], the dominant temporal modulations at slower rates are equally observed in ensemble and single instrument music[13]. In contrast, speech shows faster dominant amplitude modulations at the syllabic rate around 4–8 Hz across languages[13,15,63]. Furthermore, different rhythmic characteristics of speech and music were not only observed in the produced signals but are also reflected in the perceptual performance. For example, beat deviance detection in pure tone sequences has been shown to be maximal for beat rates of about 1.4 Hz[60]. In contrast, speech comprehension performance has been suggested to be highest for syllable rates in the theta range (~4.5 Hz) and drop at faster rates around 9 Hz[64,65] (or at even higher rates[27,66]). Accordingly, on a neural level, overlapping brain areas recruited for speech and music processing[3,4,6] have been suggested to show frequency-specific selectivity for speech and music (preprint:[67]). It should be noted that besides these dominant rhythmic modulations, speech and music also contain several hierarchical levels of information with rhythmic modulations at different time scales[68]. For example, speech contains rhythmicity beyond the syllable level[20,39,42,69] at the phrasal level at around 1–2 Hz[33,34,70–75]. Music contains rhythmic fluctuations beyond the beat rate at faster single note rates or slower phrasal rates[18,76]. In summary, speech and music show characteristic rhythmic profiles and might involve partially distinct rhythmic timing mechanisms.

Speech and non-vocal music production typically employ different motor effectors, which may recruit specific parts of the motor system related to rhythmic processing. Speech is produced by the mouth (lips, tongue, jaw) and the vocal cords. Other motor effectors such as the hands and arms can additionally support non-verbal aspects of speech production. Non-vocal music production commonly relies on the hands and arms (or sometimes the feet). For singing, the mouth and the vocal cords are used, though in a different manner when compared with speech (preprint:[77,78]). Thus, the differences in rhythmic motor timing might depend on the distinct use of motor effectors when producing speech or music. Accordingly, different motor effectors have been previously related to different sensitivities for production rates in interlimb coordination, with the mouth and vocal cord being superior in precise rhythmic pattern production at fast rates compared to the arms and feet[79]. Differences related to motor effectors have also been reported in the context of spontaneous production rates. Rhythmic motor timing in music has been traditionally researched in finger-tapping paradigms[80–84]. Spontaneous finger-tapping rates have been observed around 2 Hz[60,82,83,85–87], with optimal synchronization of finger-tapping to the beat at these rates[82,83,88]. The repetition of piano melodies by trained pianists has revealed similar spontaneous rates around 2 Hz[89], which were correlated with the individual spontaneous finger-tapping rates. Fewer studies investigated spontaneous syllable production rates and found optimal rates around 4 to 8 Hz in natural speech production[27,63]. Other methods require individuals to repeatedly whisper a single syllable, and confirmed spontaneous rates around 4–5 Hz[90]. In the speech domain, structural and functional connectivity between auditory and speech-motor regions have been associated with the ability to synchronize speech perception and production at syllabic rates of about 4.5 Hz[91,92]. In these studies, perception-production synchronicity was measured using the behavioral protocol of the spontaneous speech synchronization test (SSS test)[91,93]. Using the SSS test, it was demonstrated that high synchronization strength was related to increased speech and auditory perception performance measured in various tasks[90–92,94]. Interestingly, speech perception-production synchronization and, on the neural level, auditory-motor cortex coupling seem to be strongest at syllable rates of 4.5 Hz[16,95]. Whether perception-production synchronization in music shows similar rate-restrictions, and whether synchronization is optimal at distinct rates for speech and music,

remains unclear. In summary, the specific rhythmic characteristics of the produced speech and music signal together with the distinct spontaneous production rates observed for different motor effectors may indicate domain-specific rhythmic motor timing.

In a behavioral paradigm, we tackle the question of domain-specific mechanisms by investigating whether the optimal time scales in the speech and music domain differ and depend on the motor effector involved in their production. The optimal rate was defined as the stimulus presentation rate with highest performance. In a perception-production synchronization task as well as an auditory perception task, we used speech (syllable sequences) and music stimuli (piano tone sequences) and two different motor effector systems (whispering and finger-tapping). All tasks were performed at slow rates around 2 Hz (1.92 – 2.08 Hz) and fast rates around 4.5 Hz (4.3 – 4.7 Hz). We hypothesized that specific motor effectors recruit distinct cortical rhythmic motor timing circuitry with distinct optimal processing rates that constrain the auditory-motor coupling. More specifically, we predicted that the involvement of motor effectors associated with speech is related to higher synchronization performance at fast rates around 4.5 Hz, while motor effectors associated with music show highest synchronization performance at slower rates around 2 Hz. Assuming that the corresponding motor systems are activated even without overt motor behavior in the auditory perception task[43–46], we hypothesized that the performance in the perception task should mirror the results from the synchronization task, with higher and lower rates enhancing speech and music processing, respectively. Furthermore, synchronization was expected to predict perception performance at the corresponding time scale. Alternatively, we hypothesized that rhythmic timing processes facilitated by the motor system might generally be optimal at slower time scales, which has been suggested in previous work[17,31]. This would result in higher performance at slow time scales across domains.

## Methods

The study protocol as well as the planned analyses were pre-registered on asPredicted.org (https://aspredicted.org/ci7ms.pdf) on 9 March 2022. Deviations from the preregistered procedure can be retrieved from supplementary note 2.

**Participants**. A total of 66 participants initially participated in the study. All reported being neurologically healthy, having no psychiatric disorders and having normal and uncorrected hearing. Written informed consent was obtained from all participants prior to starting the study and subjects received monetary compensation for their participation. No participants dropped out or declined the participation. All experimental procedures were ethically approved by the Ethics Council of the Max Planck Society (Nr. 2017_12). Data collection was performed from March to April 2022.

Following the procedural recommendations for the SSS test[91,93], two participants were excluded because they spoke loudly instead of whispering during the synchronization task. An additional 2 participants were excluded due to inconsistency between any two trials of the same condition in the synchronization task. Inconsistency was detected using several linear regression models predicting performance in each condition's second trial from the same condition's first trial and participants were classified as inconsistent if the performance in the second trial laid outside of the 99% confidence interval. The final sample for the synchronization task included 62 participants (36 women, 23 men, 2 non-binary, 1 undisclosed gender, age range: 18–40 years ($M = 26.28$, SD $= 4.16$). Gender was assessed by asking the participants to self-report their gender (German: "Geschlecht").

For the temporal deviation perception task, the same group of participants was tested. We excluded 4 participants due to performance at or below chance level in at least one condition (stimulus x rate). Additionally, 1 participant had to be excluded due to technical problems during data acquisition. Thus, the final sample for the perception task included 57 participants (33 women, 21 men, 2 non-binary, 1 undisclosed gender, age range: 19–40 years ($M = 26.54$, SD $= 4.12$).

**Stimuli**. To generate the tone and syllable sequences for the perception and the synchronization tasks, we used the same sets of twelve syllables or twelve piano tones, for the speech and music stimuli, respectively. For both tasks, we generated random syllable and tone sequences that resulted from randomly combining the twelve syllables or piano tones with no gap in between them. No syllables or piano tones were repeated consecutively.

All syllable sequences were created using the speech synthesizer MBROLA with a male German diphone database (de2) at 16,000 Hz. The sequences consisted of twelve distinct syllables with each syllable starting with a consonant followed by a vowel. The sequences were resampled to 44,100 Hz using the Praat software[96]. The tone sequences were generated as MIDI-files using MIDIUtil running on Python version 3.8.8. The sequences consisted of twelve piano tones (MIDI instrument number 1) and included all notes between C3 and B3 (midi notes 48 – 59). The MIDI-files were then synthesized to wav files on a high-quality soundfont using FluidSynth version 2.2.4. All stimuli were synthesized at their respective rate, based on the syllable and tone duration information provided to the synthesizer.

**Procedure**. The stimulus presentation and response recording was performed on a Windows PC and managed with the Psychophysics Toolbox Version 3.0.12[97,98] running on MATLAB version R2021a. The session took 90 min and included, in the following order: the auditory perception task, the perception-production synchronization task, and questionnaires concerning demographics and musical experience including the German version of the Goldsmiths Musical Sophistication Index (Gold-MSI[99,100]). Schematic representations of the perception-production synchronization task and the auditory perception task are displayed in Fig. 1. All auditory stimuli were presented binaurally using Ethymotic Research (ER) 3c in-ear headphones with E-A-RLINK foam eartips attached to them.

*Perception-production synchronization task*. To measure the participants' ability to synchronize their speech and music production to rhythmic sequences of piano tones and syllables, we used several adapted versions of the accelerated version of the SSS test[91,93]. While participants listened to accelerating sequences of piano tones or syllables in fast or slow rates, they were instructed to whisper or to tap in synchrony with the sequences. The order of the motor effectors (tapping versus whispering) as well as the order of the stimulus types (syllables versus piano tones) within each articulator block was randomized. Participants were instructed to tap on the table with their dominant hand within a highlighted area 3 cm around a microphone. In the whispering conditions, participants were instructed to repeatedly whisper the syllable "TEH". The whispering was recorded using a Shure MX418 Microflex directional gooseneck condenser microphone that participants placed at around 3 cm distance from their mouth. We used an audiocard (RME Fireface UC) with high precision and presented stimuli using the full duplex mode implemented in the Psychophysics Toolbox[97,98]. This mode supports simultaneous sound presentation and multi-channel audio capture without any temporal jitter. We recorded the

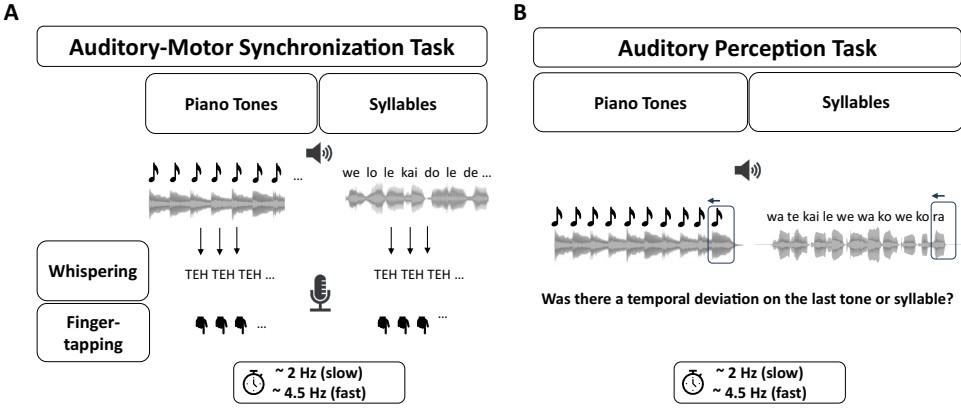

**Fig. 1 Procedure for Both Tasks. A** Procedure for the synchronization task. All participants ($n = 62$) completed both motor effector conditions (tapping and whispering) and all four stimulus x rate combinations in random order. Each articulator block started with a loudness adjustment where participants were advised to adjust the loudness level until they were no longer able to hear their own whispering or tapping. The following block included priming and synchronization sequences that were completed twice for each condition. **B** Procedure for the auditory perception task. Participants ($n = 57$) listened to sequences of syllables and piano tones at fast and slow rates. In 50% of the sequences, the last syllable or piano tone was shifted to be presented earlier and participants were asked to detect these temporal deviations.

presented stimulus with a loopback microphone, which enabled us to simultaneously record the stimulus and the participant's tapping and whispering. The complete flow for the synchronization task is shown in Fig. 1A.

Each articulator block began with a volume adjustment in which participants were instructed to adjust the volume of a syllable or tone sequence until they were not able to hear their own whispering or tapping by key presses. Depending on the condition that the subjects started with, they were played a random syllable or tone stream in the same rate as the respective start condition. The maximum amplitude was fixed at a sound pressure of 90 dB SPL to prevent hearing damage.

After the volume adjustment, subjects were first primed twice at 4.5 Hz for the fast sequences and at 2 Hz for the slow sequences. In the priming phase, subjects were first presented with a syllable or tone sequence at a given rate for ten seconds and then they were instructed to whisper or tap at the same rate for ten seconds after the audio stopped. The priming was performed with a male voice repeatedly articulating the syllable 'TEH' for the syllable conditions. When subjects were synchronizing to a tone sequence, priming was performed with a sequence of the note C1 (32.7 Hz) played on an acoustic grand piano generated as described above.

The synchronization sequences consisted of slightly accelerating tone or syllable sequences presented at fast or slow rates. This follows the established procedure of the explicit version of the SSS test[93]. Accelerating sequences are used to test for participants' spontaneous auditory-motor synchronization to slight, undetectable changes in the rate of the stimuli. The rate in the fast sequences ranged from 4.3 to 4.7 Hz and increased in steps of 0.1 Hz every 48 syllables. In the slow sequences, the rate was increased from 1.92 Hz to 2.08 Hz in steps of 0.04 Hz, accordingly. All sequences contained 240 syllables or piano tones and the length of the synchronization sequences was 50 seconds for the fast sequences and 120 seconds for the slow sequences. Subjects were asked to tap or whisper in synchrony to the sequences while listening. Participants performed two runs consisting of two priming trials and one synchronization trail for each condition.

*Auditory perception task.* The auditory perception task required participants to identify small rhythmic deviations in sequences of syllables and piano tones. Figure 1B illustrates the procedure for the auditory perception task. Each sequence consisted of ten

piano tones or syllables presented isochronously and participants were presented with a total of 80 sequences for each stimulus (syllables versus piano tones). In 50% of the trials, the last piano tone or syllable occurred early relative to the isochronous rhythm of the preceding context. For syllables, the deviation was 28–34% of the inter-onset interval, and for piano tone sequences, the deviation was 12–18% of the inter-onset interval. These percentages were obtained based on pilot testing aiming to reach a similar mean performance in syllables and piano tones across participants. The sequences were presented at fast and slow rates corresponding to the rates used in the synchronization task. The fast rates varied randomly across trials between 4.3 and 4.7 Hz in steps of 2% and the slow rates, accordingly, between 1.92 and 2.08 Hz in steps of 2%.

The stimuli were presented in two blocks for each stimulus type. The order of the stimulus blocks was randomized, as well as the order of the stimuli within each stimulus block. Fast and slow stimuli were presented randomly within each block. The sequences were presented at a sound pressure level of ~70 dB SPL. Prior to starting the first block of each stimulus type, participants received a training including feedback to become familiar with the stimuli.

**Data analysis.** The calculation of the phase-locking values (PLVs) as well as the baseline correction were performed using Matlab version 9.9.0.1592791 (R2020b). The statistical analyses were performed using R version 4.0.5 running on RStudio version 1.4.1106. The analyses relied on the packages lme4 version 1.1-28, lmerTest version 3.1-3, psych version 2.3.9, car version 3.1-0, emmeans version 1.7.2, DHARMa version 0.4.6, effectsize version 0.8.6, performance version 0.10.5, MVN version 5.9. The plots were created using ggplot2 version 3.4.4, sjPlot version 2.8.15 as well as introdataviz version 0.0.0.9003.

*Phase-locking value.* In the synchronization task, the synchronization strength between the envelope of the acoustic signal and the envelope of the motor output was measured using the PLV between both signals (with 1 denoting strong synchronization and 0 no synchronization). The PLV is calculated as described in the equation:

$$\text{PLV} = \frac{1}{T} \left| \sum_{t=1}^{T} e^{i(\theta_1(t) - \theta_2(t))} \right| \qquad (1)$$

with $t$ being the discretized time, $T$ being the total number of time points and $\theta_1$ and $\theta_2$ being the phase of the motor and the auditory signal.

The acoustic and motor envelopes were computed using the Neural Systems Laboratory (NSL) Auditory Model toolbox for MATLAB (http://nsl.isr.umd.edu/downloads.html). To extract the acoustic envelope, we applied cochlear filtering in parts of the signal between 180 Hz and 7,246 Hz. Acoustic and motor envelopes were resampled at 100 Hz and filtered depending on the rate of the stimulus. For the fast sequences, filtering was applied between 3.5 and 5.5 Hz, following the procedure reported for the SSS test[91,93]. For the slow sequences, the envelopes were filtered between 1.56 and 2.44 Hz. The phases were then extracted from the envelopes using the Hilbert transform. The PLV was calculated in windows of 5 seconds with 2 seconds overlap for the fast conditions and in windows of 11 seconds and 4.5 seconds overlap for the slow conditions. Therefore, we adjusted codes provided by Lizcano-Cortés et al.[93] available at https://doi.org/10.5281/zenodo.6142988. The PLVs for one block were estimated by averaging the PLVs for all time windows within this block.

*PLV normalization.* The tapping and the whispering signals displayed considerable differences in their acoustic properties such as differences in their amplitude. Additionally, although all sequences had the same number of cycles, the length of the fast and slow sequences differed vastly, which could possibly have an effect on the PLVs. To correct for these effects, we normalized the PLVs with respect to a permutation distribution. The permutation distribution measure was estimated by partitioning the acoustic envelope into 5 s windows for fast conditions and 11 s windows for slow conditions, respectively. These segments were then randomly shuffled and PLVs of the permutation distribution were computed using the unshuffled motor stimulus and the shuffled auditory stimulus. The baselined PLVs were finally obtained by subtracting the PLVs of the permutation distribution from the PLVs obtained using the unshuffled stimuli as explained above. To retrieve one PLV for each condition and subject, the PLVs from both synchronization trials were averaged.

*Analyses for the synchronization task.* To assess the influence of stimulus, rate, and articulator on synchronization performance, we applied a linear mixed model (LMM) with the PLV as the dependent variable. The model included a random intercept for participants to consider the 8 repeated measurements for every subject. Additionally, we included characteristics of the motor and acoustic envelopes. We hereby controlled for differences between the recorded tapping and whispering signals (motor envelope) and differences between the presented tone and syllables sequences (acoustic envelope). After calculating the absolute fourier transform of the envelopes, we identified the peak amplitude across all frequencies below 10 Hz as well as the width of the amplitude peak for every trial using the Matlab function "findpeaks". The width of the strongest peak was calculated based on the full-width half maximum. These two measures were included in the potential predictors in the mixed effects model for the synchronization task.

Predictors and random slopes were chosen by a forward stepwise regression procedure using likelihood-ratio tests. A criterion of $\alpha = 0.05$ was applied to determine if predictors should be included in the model. Potential predictors included the articulator (tapping versus whispering), the stimulus type (tone versus syllable), and the rate (fast versus slow), which were manipulated within subjects. Approximated $R^2$ values were calculated by the method suggested by Nakagawa and Schielzeth[101] yielding estimates for the explained variance when

only considering fixed effects (marginal $R^2$) as well as when considering fixed and random effects (conditional $R^2$). Effect sizes were obtained using the effectsize package in R[102]. The partial $\eta^2$ estimates provide information on the amount of evidence explained by each factor. Degrees of freedom were approximated using the Kenward-Roger method[103].

The final linear mixed effects model configuration for the synchronization task included the predictors tempo, articulator, and stimulus. Additionally, the two-way interaction between rate and articulator explained additional variance and was thus included in the final model configuration. Furthermore, the width of the peaks in the motor envelopes was included in the synchronization model. The characteristics of the acoustic envelope did not explain a significant share of variance and were therefore not included in the model. We added a random slope for the rate. No further random slopes were added into the model, as adding further random slopes led to the fit being singular. The approximate $R^2$ revealed an explained variance of $R^2 = 46.5\%$ when only considering the fixed effects. When additionally considering the random effects, the explained variance increased to $R^2 = 71.6\%$. We calculated post-hoc pairwise comparisons using the R package emmeans[104]. For the post-hoc pairwise comparisons, Kenward-Roger approximation was used for approximating the degrees of freedom and $p$ values were adjusted for multiple comparisons using the Tukey method. The resulting residuals met the normality assumption based on visual inspection and based on the Shapiro-Wilk normality test ($W = 0.997$, $p = 0.56$). We revealed a null result in one post-hoc comparison concerning the difference between tapping and whispering at fast rates. Therefore, we calculated a Bayes factor ($BF_{01}$) for a Bayesian paired samples t-test using the software JASP using a Cauchy prior distribution with $r = 1/\sqrt{2}$[105]. To investigate the power of this analysis, we conducted post-hoc design calculations based on Monte Carlo simulations using the R package BFDA[106].

In order to access the structure of dependencies between conditions in synchronization ability, we conducted a PCA using the psych package running on R. The PCA aimed at summarizing the information from the individual normalized PLVs in all eight synchronization conditions in a small number of principal components while retaining a sufficiently high share of the variance in synchronization performance. These components result from linear combinations of the observed variables (i.e., the PLVs of each participant in the eight synchronization conditions). We chose to extract 3 principal components. The number of components was chosen based on the Kaiser-Guttman criterion as well as on the visual inspection of the scree plot. According to the Kaiser-Guttman criterion, all components that display eigen values exceeding 1 are selected[107,108]. The extracted components explain a share of 70% of the variance, which conforms with commonly used criteria for the amount of retained variance[109]. The components were rotated orthogonally using varimax rotation to improve the interpretability of the components. The data met assumptions of multivariate normality based on a Henze-Zirkler test for multivariate normality ($HZ = 0.95$, $p = 0.35$). This implies that the extracted components can be regarded as uncorrelated and independent[110]. Based on the pattern of loadings (i.e., reflecting correlations) of the synchronization conditions on the rotated components, component labels were assigned. Component labels denote the synchronization conditions that showed the highest loading and therefore are simplifications of the complex dependencies.

*Analyses for the auditory perception task.* To assess the influence of the rate and the stimulus category (syllables versus piano

**Table 1 Results of the linear mixed effects model for the synchronization task**

| Effect | Estimate | 95%CI | Partial $\eta^2$ | df | P value |
|---|---|---|---|---|---|
| Intercept | 0.14 | [0.11, 0.17] | – | 87.04 | <0.001 ** |
| Rate (slow) | 0.22 | [0.19, 0.25] | 0.74 | 112.56 | <0.001 ** |
| Motor effector (whisper) | 0.01 | [−0.01, 0.03] | 0.06 | 368.49 | 0.4329 |
| Stimulus (tones) | 0.06 | [0.04, 0.07] | 0.14 | 371.73 | <0.001 ** |
| Rate × motor effector | −0.09 | [−0.12, −0.06] | 0.10 | 372.46 | <0.001 ** |
| Motor envelope width | 0.03 | [0.02, 0.04] | 0.05 | 443.91 | <0.001 ** |

$N = 62$ participants
**$p < 0.01$

tones) on perception performance, we applied a generalized linear mixed model (GLMM) with the accuracy in every trial as the dependent variable. Additionally, characteristics of the acoustic envelopes were included as potential predictors for perception performance. These characteristics were calculated analogously to the amplitude and widths used in the synchronization task.

Predictors were chosen using a forward stepwise regression procedure using likelihood-ratio tests. A criterion of $\alpha = 0.05$ was applied to determine if predictors should be included in the model. Potential predictors included the stimulus category (tone versus syllable) and time scale (fast versus slow) which were manipulated on the item level, i.e., within subjects and between items. Additionally, since we expected the synchronization performance to influence the perception performance, we included the principal components from the PCA analysis of the synchronization data as potential predictors on the subject level. We chose to include the principal components instead of the eight PLVs as predictors of the performance to avoid multicollinearity in the regression model due to medium to high correlations between the PLVs in several conditions. Finally, the relevance of interactions between all predictors was determined by the stepwise regression procedure.

The model included random intercepts for subject and stimulus to take the hierarchical structure of the data into account. The recommendations by Barr et al.[111] suggest that random slopes should be included on the subject level for within-subject predictors with several observations and their interactions. Therefore, we added random slopes for the item-level predictors rate and stimulus type and their interaction on the subject level and tested them using the stepwise regression procedure. Approximated $R^2$-values were calculated by the method suggested by Nakagawa and Schielzeth[101]. Effect sizes were calculated using the odds ratios of the parameters obtained in the logistic mixed effects model.

The final model configuration revealed by the stepwise regression procedure included rate and stimulus category as predictors on the trial level, as well as their interaction. The width of the peaks in the acoustic envelopes explained a sufficient share of incremental variance and was therefore included in the model. Additionally, the PCA components 1 (fast component) and 3 (slow tapping component) were included in the model. No interactions between rate or stimulus and the synchronization components or their three-way interactions explained additional variance (all $p > 0.05$). Additionally, including PCA component 2 (slow whispering component) did not yield an improved model fit ($X^2(1) = 1.18$, $p = 0.28$). The model included random slopes on the subject level for rate, stimulus, as well as their interaction. When only considering the fixed effects, the model accounted only for a small amount of variance in accuracy ($R^2 = 7.7\%$). When additionally considering the variance explained by the random effects, the explained variance increased to $R^2 = 24.2\%$. We calculated post-hoc pairwise comparisons using the R

package emmeans[104]. For the post-hoc pairwise comparisons, $p$ values were adjusted for multiple comparisons using the Tukey method. Model diagnostics concerning the distribution of the residuals were conducted using the DHARMa package[112], which revealed no significant deviation of the distribution of the observed residuals from the expected distribution.

**Reporting summary**. Further information on research design is available in the Nature Portfolio Reporting Summary linked to this article.

## Results

We first report the results for the synchronization task ($N = 62$) resulting from the linear mixed model predicting synchronization performance from the rate, the motor effector (whispering versus tapping), and the stimulus type. Additionally, we describe the results from the principal component analysis of the synchronization task. Finally, we report the results for the perception task, where we used a generalized linear mixed effects model to predict the accuracy from the rate and the stimulus type (syllables versus piano tones), as well as the principal components from the synchronization task.

**Effector-specific differences in synchronization**. Results for the LMM predicting synchronization performance from rate, motor effector, and stimulus type are displayed in Table 1. The LMM revealed significant main effects of rate and stimulus type, as well as a two-way interaction between rate and motor effector. Additionally, the model included a random intercept for participant as well as a random slope for rate. As we were interested in endogenous rhythmic timing mechanisms that are not reflecting processing advantages related to acoustic signal differences, we controlled for acoustic envelope characteristics in the model. We therefore added characteristics of the envelope of the speech and music signal (acoustic envelope) and of the envelope of the recorded whispering and tapping signal (motor envelope) as predictors. Characteristics of the acoustic envelope provide crucial landmarks for the neural tracking of speech and music[18,20,113], and may contribute to the perception of stimuli as speech or music[114,115] (preprint:[77,116]). The recorded whispering and tapping signals (motor envelope) might differ and confound the synchronization measure. The step-wise regression procedure revealed that only the motor envelope characteristics significantly improved the model fit and thus explained variance in synchronization performance. The acoustic envelope characteristics were therefore not included in the model. Descriptively, the motor envelope peak width was larger for tapping compared to whispering and for slow rates compared to fast rates. A larger envelope peak width was related to higher synchronization performance (Estimate (443.91) = 0.03, $p < 0.001$, Partial $\eta^2 = 0.05$, 95%CI = [0.02, 0.04]), which indicates an improved synchronization to the accelerating rhythmic structure.

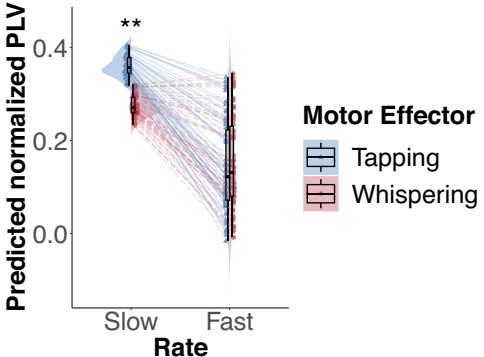

**Fig. 2 Interaction between rate and motor effector in the synchronization task.** Linear mixed model estimates for the normalized phase locking values (PLVs) and their distributions for tapping (blue triangles, solid lines) and whispering (red circles, dashed lines). The boxes display the median along with the inter-quartile range. The whiskers extend to data points within 1.5 times the inter-quartile range. Each individual point shows the model estimate for one participant ($n = 62$). The lines show the estimates of the random slope for the rate. ** $p < 0.01$.

Synchronization was better when synchronizing to piano tones than to syllables (Estimate(371.65) = 0.06, $p < 0.001$, Partial $\eta^2 = 0.14$, 95%CI = [0.07, 0.09]). Post-hoc comparisons revealed that subjects generally synchronized better at slow rates than at fast rates (Contrast fast versus slow (tapping): Estimate(113) = $-0.22$, $p < 0.001$, Cohen's d = $-2.68$, 95%CI = [$-0.25$, $-0.19$], Contrast fast versus slow (whispering): Estimate(105) = $-0.12$, $p < 0.001$), Cohen's d = $-1.53$, 95%CI = [$-0.15$, $-0.09$]), indicating that synchronizing was easier at slow rates irrespective of the domain. Subjects synchronized better when tapping than when whispering, but only at slow rates (Contrast tapping versus whispering (slow): Estimate(383) = 0.09, $p < 0.001$, Cohen's d = 1.05, 95%CI = [0.06, 0.11]). In contrast, we observed no significant differences between tapping and whispering at fast rates (Contrast tapping versus whispering (fast): estimate(369)= $-0.01$, $p = 0.432$, Cohen's d = $-0.10$, 95%CI = [$-0.03$, 0.01]). To support this result, we calculated a Bayes factor ($BF_{01}$) for a Bayesian paired samples t-test. The Bayes factor reflects the probability of the data under H0 relative to H1[117]. In this case, H0 reflected no difference between the conditions, whereas H1 reflected a difference between tapping and whispering at fast rates. The resulting Bayes factor was $BF_{01} = 9.41$, which indicates that the data were 9.41 times more likely under H0 than under H1. Heuristically, this can be classified as moderate evidence for H0 over H1[117]. The posterior distribution had a median of $Md = 0.031$, 95%CI = [$-0.142$, 0.205]. The Bayes factor seemed to be moderately sensitive to the prior width, ranging from about 7 to 19 across a wide range of prior widths. An annotated .jasp file including the data, the input choices, and the results is available at https://osf.io/9qthr/. Post-hoc design calculations based on Monte Carlo simulations revealed that a paired, two-sided t-test with the available sample size of $n = 62$ would have provided moderate evidence ($BF_{10} > 6$) for H1 for an effect size of 1.05 in 100% of the simulations. This effect size was observed for the contrast between tapping and whispering at slow rates. Therefore, it can be assumed that the study was sufficiently powered to detect effects in this range. Using an effect size of 0.5, we revealed that our study would have provided moderate evidence for H1 ($BF_{10} > 6$) in 84.7% of the stimulations and inconclusive evidence in the remaining 15.3% of the simulations. For small effect sizes approximating 0.2, only 10.5% of the simulations revealed moderate evidence for H1. Therefore, our study does not seem to be sufficiently powered to detect effect sizes in this range.

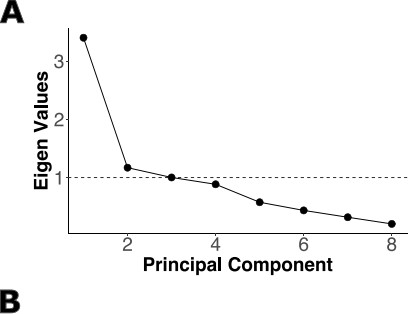

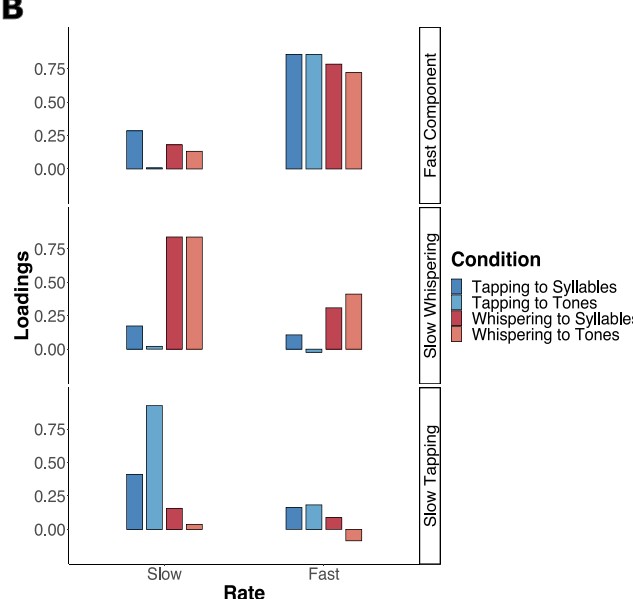

**Fig. 3 Results of the principal component analysis of the synchronization data. A** Scree Plot. **B** Loadings of all Variables on the 3 Principal Components. Loadings for the tapping conditions are displayed in blue (dark blue for tapping to syllables, light blue for tapping to tones) and loadings for the whispering conditions are displayed in red (dark red for whispering to syllables, light red for whispering to tones). The PCA is based on $n = 62$.

The interaction is illustrated in Fig. 2 and in supplementary Fig. 1. Although synchronization was generally enhanced at slow time scales, there seem to be motor effector-specific factors contributing to distinct performance profiles in speech and music. Crucially, tapping synchronization was only advantaged compared with whispering at slow rates, which is consistent with research indicating that the music production system is optimized at slower rates than the speech production system[13].

**Three distinct factors explaining synchronization performance.** The PCA of the synchronization data revealed 3 components with Eigenvalues above or at 1. The scree plot of the analysis is displayed in Fig. 3A. Following the Kaiser-Guttman criterion[107,108], we chose to focus on the 3 first components. The loadings of each synchronization condition on the 3 first components are shown in Fig. 3B.

Although labeling the components remains tentative, it can be concluded that component 1 (fast component) captures variance associated with all fast conditions. Thus, participants who synchronized well to fast sequences when tapping synchronized well to fast sequences when whispering, irrespective of the stimulus. Component 2 was mainly related to the slow whispering conditions (slow whispering component) while component 3 captured slow tapping (slow tapping component). Thus,

**Table 2 Results of the generalized linear mixed model for the perception task**

| Fixed effects | Estimate | 95%CI | Odds Ratio | P value | | |
|---|---|---|---|---|---|---|
| Intercept | 1.92 | [1.70, 2.14] | 6.80 | <0.001 ** | | |
| Rate (slow) | −0.23 | [−0.50, 0.04] | 0.79 | 0.089 | | |
| Stimulus (tones) | −0.65 | [−0.90, −0.40] | 0.52 | <0.001 ** | | |
| Rate × stimulus | 1.29 | [0.86, 1.73] | 3.64 | <0.001 ** | | |
| Acoustic envelope width | −0.20 | [−0.28, −0.13] | 0.82 | <0.001 ** | | |
| Fast component | 0.36 | [0.20, 0.51] | 1.43 | <0.001 ** | | |
| Slow tapping component | 0.22 | [0.07, 0.37] | 1.24 | 0.005 ** | | |
| **Random effects** | **Variance** | **SD** | **Correlation** | | | |
| Participant level | | | | | | |
| Intercept | 0.32 | 0.57 | – | – | – | |
| Tempo (slow) | 0.56 | 0.75 | −0.48 | – | – | |
| Stimulus (tones) | 0.43 | 0.66 | −0.53 | 0.77 | – | |
| Tempo × stimulus | 1.79 | 1.34 | 0.66 | −0.80 | −0.86 | |
| Stimulus level | | | | | | |
| Intercept | 0.18 | 0.43 | | | | |

*N* = 57 participants
\*\**p* < 0.01

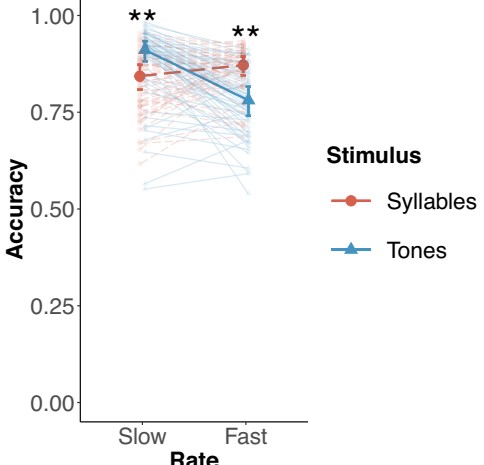

**Fig. 4 Interaction between tempo and stimulus in the perception task.** Logistic mixed effects model estimates for perception accuracy for tones (blue triangles, solid lines) and syllables (red circles, dashed lines). The highlighted lines show the parameter estimates across participants. The transparent lines show the accuracy estimates as well as the slope estimates for each participant (*n* = 57) when allowing for random slopes. The error bars show the 95% confidence intervals around the model estimates. \*\**p* < 0.01.

participants who synchronized well to slow sequences when whispering did not necessarily synchronize well to slow sequences when tapping. To summarize, the PCA results indicate that there exists a motor effector general synchronization factor at fast rates, while synchronization performance at slow rates seems to be driven by motor effector specific influences. It should however be noted that this conclusion is tentative given the small eigen values for components 2 and 3.

**Superior perception at domain-specific rates**. Table 2 provides a summary of the results for the GLMM for the auditory perception task. The results revealed a significant interaction between rate and stimulus as well as a fixed effect of stimulus. The interaction as well as the slopes and intercepts for the individual participants are visualized in Fig. 4. We conducted pairwise post-hoc comparisons to determine the direction of the interaction. As

predicted by our hypothesis, we found that syllable perception was superior compared to tone perception at fast rates (Contrast syllables versus tones (fast rates): Estimate = 0.65, *p* < 0.001, Cohen's *d* = 7.98, 95%CI = [0.40, 0.90]) whereas at slow rates, tone perception was superior compared to syllable perception (Contrast syllables versus tones (slow rates): Estimate = −0.65, *p* < 0.001, Cohen's *d* = −7.96, 95%CI = [−0.95, −0.34]). These strong effects suggest that speech and music perception seem to activate rate-specific processes with the rate preferences matching the dominant rates in the motor domain.

The results additionally revealed a significant effect of the width of the peaks of the acoustic stimulus envelope on perception performance (Estimate = −0.20, *p* < 0.001, OR = 0.82, 95%CI = [−0.28, −0.13]). Thus, characteristics of the stimulus envelope influenced perception performance, with a smaller peak width being related to higher performance. Descriptively, the peak widths were larger for the syllable sequences than for the piano tone sequences, which could be expected given the acoustic characteristics of piano tones compared to syllables. All other effects persisted after controlling for the acoustic envelope characteristics. Therefore, the rate-specific effects on speech and music perception do not seem to reflect performance differences due to acoustic characteristics of the envelope.

**Synchronization performance influences perception performance**. As expected, the GLMM additionally revealed significant fixed effects of the fast synchronization component (Estimate = 0.36, *p* < 0.001, OR = 1.43, 95%CI = [0.20, 0.51]) and the slow tapping component (Estimate = 0.22, *p* = 0.005, OR = 1.24, 95% CI = [0.07, 0.37]) indicating that perception performance was positively influenced by synchronization performance. That means that a better synchronization performance, as defined by a higher PLV, predicted higher auditory perception performance. This suggests a link between motor and perceptual performance and is consistent with previous work emphasizing the importance of motor contributions to perceptual performance in the auditory domain. The slow whispering component did not explain a significant share of variance in the step-wise regression procedure and it was therefore not included in the model.

**Control of potential confounds**. To ensure that the effects revealed by our analyses are not merely an artifact of characteristics of the stimuli, the experimental procedure, or further

confounding factors, we conducted a series of control analyses. The control analyses suggest that the synchronization performance is not influenced by the order of the experimental conditions, indicating that no practice or fatigue effects were significantly affecting the synchronization performance (see supplementary table 1). Additionally, we revealed that self-reported musical sophistication influenced synchronization performance, but all other effects remained constant when controlling for musical sophistication (see supplementary table 2). Musical sophistication was correlated with the fast component and the slow whispering component (see supplementary note 1). However, performance in the synchronization task, as reflected in the PCA components, predicted perception accuracy beyond effects of musical sophistication (see supplementary table 3).

## Discussion

Speech and music display similarities but also characteristic differences in their temporal structure. Yet, it is unclear whether distinct rhythmic timing mechanisms are recruited in the speech and in the music domain. The results presented here provide insights into rate-specific processing for perception and synchronization in both domains. In an auditory perception task, duration discrimination in piano tone sequences was highest at slower rates of around 2 Hz, whereas it was highest at faster rates around 4.5 Hz for syllable sequences. These time scales correspond to the previously described dominant acoustic rhythms for produced music and speech, respectively. Regarding the auditory-motor synchronization task, the picture was more complex. We observed that synchronization was overall better at slower rates when compared with faster rates. Crucially, the synchronization performance for the different motor effectors associated with speech and music varied depending on the rate. At slow rates, finger-tapping synchronization was better compared to whispering synchronization and synchronization was related to two independent components. In contrast, at fast rates, no differences between finger-tapping and whispering synchronization performance were observed, which were related to one component reflecting dependent processes. This suggests partially distinct rate-specific processes, with independent rhythmic timing mechanisms for different motor effectors at slow but not at fast rates.

The perception task clearly indicates that the perception of syllable and piano tone sequences shows highest performance at different time scales (Fig. 4). The detection of small temporal deviations in syllable sequences was superior at faster rates of around 4.5 Hz. In contrast, deviations in piano tone sequences were detected better at slower rates around 2 Hz. The findings are consistent with previous research indicating that produced speech signals exhibit dominant temporal modulations at faster rates than music signals[13,15,118] and that these rates are reflected in optimal perception performance[14,27,60,64,66]. A possible interpretation is that speech and music signals activate cortical rhythmic timing circuits with different optimal rates, resulting in better processing at these rates. On the neural level, such optimal processing rates have been related to preferred auditory and motor cortex brain rhythms in the same frequency range[16,31]. Syllable processing has been particularly linked to faster theta brain rhythms in the auditory cortex[16,20,39,42] and speech motor areas (inferior frontal gyrus)[42,91,119], and musical beat processing to slower delta brain rhythms in the supplementary motor area[17,51,53,54].

The results of the production-perception synchronization task only partially support our hypothesis concerning different optimal time scales in music and speech processing (Fig. 2). The overall advantage of slow time scales (mixed effects model)

suggests that synchronization was highest around 2 Hz irrespective of the involved motor effector system or domain. This is consistent with behavioral findings indicating spontaneous production rates for finger-tapping or marching around 1–2 Hz[60,82,83,85–87,120], and neural findings that suggest slow delta brain rhythms in the motor cortex constrain rhythmic motor timing and render it optimal at these rates[29,31]. Additionally, the interaction between rate and motor effector reveals that, at 2 Hz, synchronization performance was better for tapping compared to whispering, whereas performance did not differ at 4.5 Hz. Partially in line with our hypothesis, this might suggest that motor effectors typically associated with music (i.e., the fingers) recruit rhythmic motor timing that is optimal at slow rates. Although synchronization performance for motor effectors associated with speech (i.e., mouth and vocal cord) remains challenging at fast rates, finger-tapping synchronization showed no advantage compared to whispering at fast rates. Alternatively, the observed effects could result from peripheral constraints for fast finger movements. The advantage of finger-tapping compared to whispering might be only present at slow but not at fast rates because of constraints that reduce the accuracy of synchronized finger-tapping at fast rates. However, peripheral constraints cannot account for our findings in the perception task in which no overt production was required. We therefore suggest that the findings reflect the recruitment of higher-level rhythmic motor timing in speech and music rather than, or in addition to, differences in peripheral muscle movements. Despite their high significance, it should be noted that the magnitude of the effects in the synchronization task was rather small. Additionally, the results did not reveal any interaction between stimulus type and the motor effector or the rate, which we expected based on the close association of stimulus types and motor effectors. Interestingly, we show the expected interaction of stimulus type and rate in the perception task, indicating that the syllable and piano tone sequences did indeed activate the respective rhythmic timing mechanisms. A possibility is that the fixed effect of the stimulus type dominated in the synchronization task, as synchronization performance was overall higher for piano tones compared to syllables across conditions. In the perception task, we controlled for an overall effect of stimulus type by matching the task difficulty across conditions.

The PCA results provide further insights by indicating that domain-specific processes, with independent patterns for the different motor effectors, are operating at slow time scales (Fig. 3). Although the results from the mixed effects model indicate that overall synchronization was better at slow rates, the PCA revealed no evidence that this reflects domain-general processes shared across motor effectors. Visual inspection of the mixed model predictions (Fig. 2) shows tight non-overlapping distributions for the synchronization of finger-tapping and whispering at slow rates. In contrast, the distributions were overlapping at fast rates. Accordingly, at fast rates, individuals with better whispering synchronization performance also showed better finger-tapping performance, resulting in one PCA component. This tentatively suggests that there exist domain-general influences that drive synchronization ability at fast rates. Our findings are in line with a very recent study that compared clapping and whispering synchronization at fast rates around 4.5 Hz and found similar performance across motor effectors[84]. Furthermore, a common mechanism for the neural tracking of speech and music at faster rates has been suggested (with other findings of this study, however, being in contrast to ours and direct comparisons being hindered because of broader frequency ranges and other methodological differences)[76]. Vocal music may provide an interesting case for future research. Speech and song overlap with regard to their motor effectors, while song shows

acoustic characteristics similar to that of non-vocal music[114,115], (preprint:[116]). This has been related to a different engagement of the motor effectors. Therefore, we expect singing synchronization to recruit rhythmic motor timing associated with the music domain that is optimal at slow time scales. To summarize, our findings from the synchronization task provide support for distinct rhythmic motor timing across motor effectors associated with speech and music processing at slow rates and overlapping mechanisms at fast rates. Previously, the behavioral performance in speech perception-production synchronization at about 4.5 Hz has been shown to correlate with the functional and structural auditory-motor cortex coupling strength[91]. Our findings suggest several distinct cortical coupling mechanisms, that is, auditory-motor coupling at about 4.5 Hz is expected to be independent of that at 2 Hz, while the latter can be assumed to differ for different motor effectors. Studies using electrophysiological measures may be able to test this prediction and further enlighten the neural substrates underlying the rate restrictions observed in our behavioral protocol

The overall perception performance across rates was most strongly predicted by the synchronization ability at fast time scales (fast PCA component). This is consistent with previous studies that associated high synchronization performance in the SSS test with increased syllable discrimination performance at fast and slow rates[90] (however, see ref. [94]). Additionally, performance in the slow tapping conditions (slow tapping PCA component) was predictive of perception performance across rates and modalities, while the performance in the slow whispering conditions (slow whispering component) was not predictive of the perception performance. Interestingly, we found that only the fast synchronization PCA component – that generalized across motor effectors – was highly correlated with musical sophistication (supplementary note 1). Thus, musical training might relate to the common influence driving synchronization ability at fast rates independent of the motor effector system. This is consistent with previous results indicating an association between musical sophistication and synchronization at fast rates in the speech domain[91,121].

**Limitations**. Our study has a limited scope in stimulus material and motor effector choice (i.e., syllable and piano tone sequences instead of natural speech and music and whispering and finger-tapping instead of natural speech and music production). However, the benefit is that our speech and music conditions are well-matched acoustically, and we show that our results are not merely caused by differences in the acoustics. We refrained from using more complex stimulus material in order to enable a close matching of the syllable and piano tone sequences. However, investigating how additional contextual information affects optimal processing rates in perception and production requires future research. Additionally, a potential limitation of our work is the use of whispering instead of natural speaking in the synchronization task. The rationale behind this decision – following the protocols of the SSS test[91,93] – was that auditory feedback from one's own speech production was minimized by the low tone of voice. As whispering involves the mouth and vocal cords in a very similar manner as speaking (while the vocal cords are not vibrating), we would not expect differences in motor effector associated rhythmic timing[91]. Findings from the perception task, in which spoken syllables (no whispering) were used, are in line with this assumption. Our findings do not aim to speak towards the minimal acoustic features that are required to elicit speech or music-specific processing, which have been researched elsewhere[114,115,122], (preprint:[77,116]). Concerning the absence of a difference between tapping and whispering at fast rates in the

synchronization task, we observed that the study was not sufficiently powered to detect small effect sizes based on post-hoc Monte Carlo simulations. However, given all other effect sizes in the post-hoc comparisons in our study were large, we do not assume that these small effect sizes are theoretically meaningful in our domain.

In conclusion, we show that discrimination of temporal deviants versus regular occurrences at faster rates was better in syllable sequences compared to tone sequences and the opposite was the case for slower rates. Our analysis of auditory-motor synchronization revealed that although performance was overall higher at slow rates, synchronization at slow rates was related to independent principal components for different motor effectors associated with speech and music. In contrast, synchronization at fast rates was correlated across motor effectors of the speech and music domain. This suggests partially distinct and partially overlapping rhythmic timing mechanisms - associated with the motor effectors - seem to be involved in music and speech processing.

## Data availability
The anonymized data including responses in the perception task as well as questionnaire responses have been deposited at https://osf.io/9qthr/. Additionally, the repository contains the baseline corrected PLVs. Raw audio recordings cannot be provided for data protection reasons, instead we provide them as processed data (i.e., envelopes).

## Code availability
The custom analysis code used to conduct the analysis is available at: https://osf.io/9qthr/. The analyses were conducted using MATLAB version R2020b and R version 4.0.5 running in R studio version 2023.09.1 + 494.

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

## Acknowledgements
We thank the German Academic Exchange Service funded by the Federal Ministry of Education and Research, as well as the Max Planck Institute for Empirical Aesthetics and the Max Planck NYU Center for Language, Music, and Emotion (CLaME) for funding this project. We thank Dr. Florencia Assaneo for very helpful discussion and Dr. Klaus Frieler for advice on the statistical analysis.

## Author contributions
A.V.B., J.M.R., C.P., and M.J.H. planned the study. A.V.B. collected the data and conducted the analysis supervised by J.M.R.; A.V.B. and J.M.R. wrote the manuscript and A.V.B., J.M.R., C.P., and M.J.H. edited the manuscript.

## Funding

## Competing interests
The authors declare no competing interests.
