## [Peer review file · Communications Psychology]

Decision letter and referee reports: first round

29th Mar 23

Dear Ms Barchet,

Your manuscript titled "Auditory-Motor Synchronization and Perception Suggest Partially Distinct Time Scales in Speech and Music" has now been seen by 2 reviewers, whose comments are appended below. We apologise for the delay in processing your manuscript, which was due to reviewer unavailability. Thank you for your patience.

I have discussed the reports with my colleagues and I regret to inform you that we decided that in light of the referee reports, we cannot publish your manuscript in Communications Psychology.

You will see that the reviewers raise substantive concerns. Taking these points together with our editorial considerations, these reservations preclude publication of this study in Communications Psychology.

The amount of work that would be required in a major revision to alleviate all concerns about potential confounds expressed across both reviews, which amounts to rerunning the entire study, exceeds what the journal considers a feasible revision request.

I am sorry that we cannot be more positive on this occasion and thank you for the opportunity to consider your work.

Best regards,

Antonia Eisenkoeck

Antonia Eisenkoeck
Senior Editor
Communications Psychology

REVIEWERS' EXPERTISE:

Reviewer #1: beat/rhythm perception, statistics

Reviewer #2: beat/rhythm perception, speech production, auditory-motor synchronization

REVIEWER COMMENTS:

Reviewer #1 (Remarks to the Author):

This study examines effects of rate in isochronous speech and tone sequences on performance in auditory-motor synchronisation and perceptual tasks. Independent experimental variables include (1) stimulus rate (2 Hz vs 4.5 Hz), (2) stimulus type (syllables vs tones), (3) motor output tested for synchronisation (tapping vs whispering). Dependent variable (performance measure) is the ability to (1) synchronise motor output to the stimulus, or (2) detect an irregularity in the rhythm. The underlying rationale is that rate preferences might be domain-general or -specific, and this should be visible in corresponding performance effects. The authors report various observations that point to, apart from some domain-general effects, a preference of slower (2 Hz) and faster (4.5 Hz) rates for motor outputs associated with music (tapping) and speech (whispering), respectively.

This is an excellent study that will be of interest to the scientific field and beyond. I do have some questions that should be seen as an effort to raise additional points of discussion rather than serious criticism.

Given the differences in dominant rates that the authors describe for music and speech (slower for the former), it would seem more natural to tap to a 2-Hz tone sequence than to a 4.5-Hz one, and

vice versa for whispering. I wonder if the authors' findings are driven by typical stimulus properties that the systems are *used* to process, rather than a demonstration of actual "preferred rates" that the systems are *built* to process.

"Music" is defined as a random sequence of tones, and I wonder whether and how this choice might have influenced results. Speculatively, what do the authors think would happen if they made these sequences "meaningful" (using, e.g., melodies), or used tasks centred on music or speech perception (i.e. on their interpretation rather than perception)? What do the authors consider the defining feature in the tones that has caused the domain-specific preference for slower rates – is it enough to present sounds with a clearly defined pitch (driven by a single sound frequency) to produce such music-specific effects? What type of stimuli would the authors use to contrast general auditory (rather than music-specific) effects with speech-specific ones?

Introduction: "Speech and music production typically employ distinct articulator systems". But mouth and vocal cords are enough to make music, and hands and arms are also employed during human communication. Do the authors predict different preferred rates for vocal music?

"Whispering synchronization was less disadvantaged compared with tapping at fast rates, which is consistent with research indicating that the speech system is optimized at faster rates than the music system." Could such a result not simply stem from differences in properties of muscles required (e.g., we cannot move fingers as quickly as we can move the jaw)? Unless the authors consider these to be part of specific "speech and music systems" (which might be good to state explicitly)?

"Faster beta brain rhythms around 20 Hz and slower delta brain rhythms around 2 Hz [...]". I found this sentence a bit confusing, as 2 Hz and 4.5 Hz are used for stimulus presentation in the experiment, and 20 Hz does not appear to be relevant.

I leave it to the authors, but the recent paper by Zuk et al (2021, Plos Comp Biol) might be an interesting addition to the manuscript.

Reviewer #2 (Remarks to the Author):

This study investigated rate-dependent differences between speech and music using a perception and a synchronization task. The main finding of the perception task was that tone perception was optimal at slower rates (~2 Hz), syllable perception was optimal at faster rates (~4.5 Hz). In the synchronization task, the authors found a higher PLV between auditory stimuli and both articulators at slower rates than the faster rate, and a higher PLV with the tone stimuli than the syllable stimuli. Further, significant interaction effects showed that PLV was higher when tapping than whispering, as well as synchronizing to tones than the syllables, only at slower rates rather than the faster rates. The authors concluded there is a domain-specific effect of articulator and stimuli at slower rates while a domain-general effect no matter of the articulator or stimuli at the faster rates. The study was pre-registered. Some analyses deviated from the original plan but were clearly explained in the supplementary material.

This work is interesting and it is worthwhile to examine the sources of rate-dependent effects on speech and music. It's an important question, thus the magnitude of comments below. At present there are a few major conceptual and analysis issues that must first be addressed that prevent it from reaching this journal's required bar of influencing thinking in the field. The most serious reservation is that comparing single-speaker speech modulation to ensemble music modulation is totally incommensurate. The 2Hz modulation of music is not only (and perhaps not even primarily) to do with articulator neuromuscular dynamics (especially a single finger) but is a constraint of it being an ensemble activity linked by an internalized sense of beat. This seems like a major showstopper. Less fatal, but in need of some more justification is that calling the finger the music articulator and voice (uniquely) the speech articulator seems to overly trivialize the reality of music. There are a number of additional, smaller needed clarifications, most of which all seem readily addressable.

General comments

1. The hypotheses could be more clearly stated. What are the hypotheses among tempo, stimuli, and articulator? Do the authors assume that different articulators are linked to distinct motor processing circuits, which have different rate sensitivities? If so, do they further assume multiple domain-specific time processing circuits? Please spell out the fundamental mechanistic hypotheses of this study.
2. The link between the articulators and certain stimuli, or domains, could stand to be further explained. The authors designed this study by presenting finger tapping as a "music articulator" and whispering as a "speech articulator". What is exactly is meant by a whispering articulator, since speech requires the coordination of multi-articulators such as mouth, lip, tongue, vocal cord, etc. Was whispering chosen in order to focus on a single vocal articulator? It is not clear. It would help if the authors more clearly explained the reasoning of selecting these two articulators and consider measuring the average rate and range of whispering and tapping rates to present a kind of ground truth of production abilities.
3. Unfortunately, the entire finger/music voice/speech dichotomy is less than clear! Many musical behaviors require both the mouth and fingers or the mouth and no fingers, e.g. wind instruments and singing. There is very little music that involves a single digit.
4. Most fundamentally, the comparison of differences in peak modulation frequency for speech and music seems like a red herring and an apples to oranges comparison. The 'faster' speech modulation result is for a single speaker, while the 'slower' music modulation rate is for musical ensembles with multiple instruments. These are not commensurate at all! A proper comparison, if there even is one, might be to look at the modulation rate of e.g. musical solos on instruments ideally played only with a single articulator. (In reality, instruments played with multiple digits can have a much higher frequency of AM relating to individual notes, not the 'beat rate' modulation that emerges in ensemble music.) This neglects secondary aspects of musical articulation such as accents and loudness, which occur at rates slower than the main articulation—that is something else entirely than simple SMS). As I think may have been mentioned, there is a key difference between tactus rate and beat reate. Not the same thing at all.
5. Possible solutions: There are probably some bass lines that are played with a single finger around the beat rate. Alternatively, for the speech side a better comparison would be to look at the modulation of many people chanting together. I suspect that will slow to be close to music— However, the fundamental underlying and basic point is that $\sim 2\text{Hz}$ modulation is likely a constraint of multi-person synchronization (mediated through a shared beat) and not an articulatory constraint!
6. Based on the preceding items, which made the same point in multiple ways, it becomes hard to see how the logic of this study makes sense, though am open for it to be further clarified and justified that it makes sense.
7. The Introduction thoroughly reviewed the neural evidence regarding speech and music but did not mention much on the behavioral evidence. Given that this is a behavioral study, please present more behavioral evidence background.
8. There is one analysis flaw that affects the main conclusions: The critical point that acoustic features of the stimuli had a very large influence on the results was underplayed and placed in the supplementary materials. We think that the acoustic features are essential to address in the main text. Specifically, Line 217 described the results as not being driven by the envelope characteristics, but comparing Table 2 to Supplementary Table 1 this is clearly not the case: All findings except for the Tempo effect were lessened or erased by controlling for the envelope characteristics. Based on the results, the envelope characteristics does not seem like a confound. Instead, it is likely a very important factor driving the main PLV results and therefore should be the main analysis. Onset profile effects are mentioned at line 167 in text.

Detailed comments

Line 24: This sentence is a bit confusing. How can music and speech rhythmic structure have emerged from the auditory or motor system?

Line 29: Could spell out the slow and fast rate in the Abstract for clarity.

Line 34: Not sure how the synchronization strength is defined.

Line 58: Please clarify the "rhythmic characteristics"

Line 59: Please discuss more about what specific areas in the motor system are referred to here

Line 71: Music production is not necessarily employed by arms and hands. Actually, vocal cord has

been used in many forms of music production.

Figure 1: Suggest adding "temporal" in front of the deviation

Table 2: There is no significant interaction effect between Articulator and Stimulus on PLV (as well as Supp. Table 1), is this expected by the authors? Isn't this finding against the hypothesis that certain articulator is preferred in certain domain (i.e. music vs. speech here)?

Line 132-3: This is a big assumption, please justify.

Line 139: were training/experience effects taken into account? People naturally vary a lot in SMS ability depending on if and which instrument they may play.

Line 152: Why were two different models (LMM vs GLMM used)? On my understanding, it would only make sense if a different link function was needed for GLMM.

Relatedly, the text here mentions only GLMM (presumably linear link) but the supplementary materials discuss it as a logistic model.

Line 155/Figure 1. Key methodological question: how were sounds sped up/slowed down? Were syllables stretched or only the onset times adjusted? The former would be problematic since it would alter the temporal sharpness of the onset. Please clarify.

Line 158: to be fair, these are weak interactions.

Line 162: The Results are very well written and easily to follow along with Table 2!

Table 2: However, confused as Articulator x Stimulus would seem to be the key factor in supporting the main hypothesis of the paper, yet it is not significant.

Line 174: How can we see this claim in the presented data? It is not clear how it is supported.

Figure 2 is so derived (using the difference) so it is very hard to see any such effects.

Line 185: What is the Kaiser-Guttman criterion. Please offer a citation or explanation.

Line 188: Component labels are a bit tricky especially that the "Slow Whispering" PC which also loads on slow tapping to syllables and fast whispering to tones.

Table 3: How about the Slow Whispering PC?

Line 203: Sorry to nitpick, but there is no kind of 'optimality' shown here, just things that are relatively better or worse.

Line 213: Please specify how "improved synchronization performance" and "increased auditory-motor synchronization performance" are defined. This is particularly important for making a conclusion about the correlates between synchronization and perception along the whole paper.

Line 221: Please specify which are the "results described above".

Line 235: What is the cut-off point between slow and fast rate? Maybe worth mentioning in the Introduction.

Line 257: Can the authors really claim the activation of respective motor systems with the current design? Please clarify the reasoning behind.

Line 266: Ref 13 sounds cool, and is very relevant, but it is unpublished!

Line 268: Important reference to consider, on the surface seems like it might be contradictory: Zuk, N. J., Murphy, J. W., Reilly, R. B., & Lalor, E. C. (2021). Envelope reconstruction of speech and music highlights stronger tracking of speech at low frequencies. *PLoS computational biology*, 17(9), e1009358.

Line 277: Is this claim about domain-general mechanisms at _slow time scale_ potentially contradictory the finding mentioned in Line 295? Please discuss more.

Line 283: Caution against using the terms "speech articulator" and the "music articulator" since first, the stimuli used in this study do not necessarily represent the music and speech; second, the articulators regarding these two domains are not limited to the articulators chosen by the authors in this paper.

Line 309-312: I may be dense but I don't quite see the argument here.

Line 365: It would be great to keep the terms consistent e.g. perception vs. production task or perception vs. synchronization task

Line 378: Curious to know why the authors didn't include the baseline condition such as hearing without performing and performing without hearing (this condition is important to understand the physical limitation of the articulators' movement, and assessing individual differences).

Line 396: How did the authors synchronize the recorded whispers and taps to the auditory stimuli? Please add the details to the text.

Line 409: Maybe justify why using "TEH", is it repeated in the syllables used for the main task?

Line 413: Totally missed from the above why the sequences are accelerating. Never would have expected that. Why is it done? Is it just to ensure people don't memorize some tempo?

Line 450: Would have been much better for your arguments if your bandwidths had been properly scaled! The high bandwidth should have been 4.5 Hz wide not the same 2Hz as used at slow

Decision letter and referee reports: first round

tempo.

Line 498: PCA on what? Please explain the procedure more completely.

Reviewer #4 Review Attachment #2

I am doing a joint review with my advisor Dr. John Iversen and will submit on his portal together.
Thank you.

30th May 23

Dear Ms Barchet,

Thank you for your correspondence asking us to reconsider our decision on your Article, "Auditory-Motor Synchronization and Perception Suggest Partially Distinct Time Scales in Speech and Music". After careful consideration we have decided that we would be willing to consider a revised version of your manuscript.

Your revised manuscript must include revisions that address each of the referees' original points of criticism you mentioned in your email.

This includes the issues listed in your appeal letter, i.e., inclusion and discussion of the envelope control analysis in the main manuscript; clarification of the nature of stimuli to avoid confusion over multiple vs single instrument; and further justification for choosing the finger and the voice as main articulators.

Along with your revised manuscript, you should also submit a separate point-by-point response to all of the concerns raised by the referees, in each case describing what changes have been made to the manuscript or, alternatively, if no action has been taken, providing a compelling argument for why that is the case. If we feel that a substantial attempt has been made to address the editorial concerns and referees' comments, this response will be sent back to the referees - along with the revised manuscript - so that they can judge whether their concerns have been addressed satisfactorily or otherwise.

I should stress, however, that we would be reluctant to trouble our referees again unless we thought that their comments had been addressed in full.

When revising your paper please:

- ensure that it complies with the editorial policies outlined below
- ensure it meets our format requirements as set out in our [Guide to Authors](http://www.nature.com/nathumbehav/info/gta) and highlighted below.
- ensure that the statistics reporting and interpretation is in line with journal guidelines <https://www.nature.com/commspsychol/submit/submission-guidelines#statistical-guidelines>

Please mark all correspondence via email with your Communications Psychology reference number in the subject line.

If the revision process takes significantly longer than five months, we will be happy to reconsider your paper at a later date, provided it still presents a significant contribution to the literature at that stage.

Please use the following link to submit your revised manuscript, point-by-point response to the Reviewers' comments with a list of your changes to the manuscript text (which should be in a separate document to any cover letter) and any completed checklist:

<https://mts-commspsychol.nature.com/cgi-bin/main.plex?el=A2DU7zX4A6RI1J5A9ftdDqdl8rEUx9DQDORhRLkS8QZ>

Please do not hesitate to contact me if you have any questions or would like to discuss the

required revisions further.

Best regards,

Antonia Eisenkoeck

Antonia Eisenkoeck
Senior Editor
Communications Psychology

EDITORIAL POLICIES AND FORMATTING

Editorial Policy: [Policy requirements](https://www.nature.com/documents/nr-editorial-policy-checklist.pdf) (Download the link to your computer as a PDF.)

Furthermore, please align your manuscript with our format requirements, which are summarized on the following checklist:

[Communications Psychology formatting checklist](https://www.nature.com/documents/commspsychol-style-formatting-checklist-article-rr.pdf)

and also in our style and formatting guide [Communications Psychology formatting guide](https://www.nature.com/documents/commspsychol-style-formatting-guide-accept.pdf) .

* **CODE AVAILABILITY:** All Communications Psychology manuscripts must include a section titled "Code Availability" at the end of the methods section. In the event of publication, we require that the custom analysis code supporting your conclusions is made available in a publicly accessible repository; please choose a repository that provides a DOI for the code; the link to the repository and the DOI must be included in the Code Availability statement. Publication as Supplementary Information will not suffice. We ask you to prepare and upload code at this stage, to avoid delays later on in the process.

* **DATA AVAILABILITY:**

All Communications Psychology research manuscripts must include a section titled "Data Availability" at the end of the Methods section or main text (if no Methods). More information on this policy, is available at <http://www.nature.com/authors/policies/data/data-availability-statements-data-citations.pdf>.

At a minimum the Data availability statement must explain how the data can be obtained and whether there are any restrictions on data sharing. Communications Psychology strongly endorses open sharing of data. If you do make your data openly available, please include in the statement:

- Unique identifiers (such as DOIs and hyperlinks for datasets in public repositories)
- Accession codes where appropriate
- If applicable, a statement regarding data available with restrictions

- If a dataset has a Digital Object Identifier (DOI) as its unique identifier, we strongly encourage including this in the Reference list and citing the dataset in the Data Availability Statement.

We recommend submitting the data to discipline-specific, community-recognized repositories, where possible and a list of recommended repositories is provided at <http://www.nature.com/sdata/policies/repositories>.

If a community resource is unavailable, data can be submitted to generalist repositories such as [figshare](https://figshare.com/) or [Dryad Digital Repository](http://datadryad.org/). Please provide a unique identifier for the data (for example a DOI or a permanent URL) in the data availability statement, if possible. If the repository does not provide identifiers, we encourage authors to supply the search terms that will return the data. For data that have been obtained from publicly available sources, please provide a URL and the specific data product name in the data availability statement. Data with a DOI should be further cited in the methods reference section.

Communications Psychology is committed to improving transparency in authorship. As part of our efforts in this direction, we are now requesting that all authors identified as 'corresponding author' create and link their Open Researcher and Contributor Identifier (ORCID) with their account on the Manuscript Tracking System prior to acceptance. ORCID helps the scientific community achieve unambiguous attribution of all scholarly contributions. You can create and link your ORCID from the home page of the Manuscript Tracking System by clicking on 'Modify my Springer Nature account' and following the instructions in the link below. Please also inform all co-authors that they can add their ORCIDs to their accounts and that they must do so prior to acceptance. <https://www.springernature.com/gp/researchers/orcid/orcid-for-nature-research>

Point-by-Point Responses to all issues raised by the reviewers

We thank the editors and reviewers for their time and effort, the positive evaluation and the constructive feedback. We revised the manuscript in the following way:

1. As Reviewer 1 and 2 both asked for clarifications, we largely revised the introduction and discussion sections to clarify our hypotheses and discussion about how speech and music stimuli and motor effectors might recruit rhythmic timing mechanisms in motor cortex.
2. We clarified that our stimuli are all single stream speech/music stimuli and discuss literature that suggests that the role of the beat in music ensemble coordination is not of concern for this study (responding to comments of Reviewer 2). Additionally, Fig.1 was revised to clarify this issue.
3. We now discuss a study that was published very recently in Nature Communications Biology, because we believe that our data nicely confirm and extend these results (Mares, Solana, Assaneo, 2023). Following this study, we adopted a hopefully clearer terminology by referring to what we previously called “articulators” now as “motor effectors”.
4. Following comments of Reviewer 1 and 2, we justified our choice of motor effectors associated with the speech (mouth and vocal cords, as accessed in a whispering task) and music domain (fingers, as accessed in a finger-tapping task). We revised the introduction and discussion to clarify and more cautiously phrase this issue and discuss limitations. Additionally, we reviewed the behavioral literature on the spontaneous production rates in speaking and tapping more thoroughly to support our argumentation.
5. As suggested by Reviewer 2, we included the envelope control analyses as the main analyses for the synchronization and the perception task. We adjusted the methods and the results sections accordingly and discussed the influence of the envelope characteristics.
6. As suggested by Reviewer 2, we updated Figure 2 in order to enhance clarity.
7. We updated a minor analysis flaw (pointed out by Reviewer 2) and now scaled the filtering bandwidths in the calculation of the PLV at different rates proportionally. We adjusted the methods and all results accordingly. Note that due to this change the model now converged with a random slope for the rate across participants, while none of the effects we previously reported were altered.
8. As suggested by Reviewer 1, we discussed several limitations and future directions including rate preferences in vocal music, potential influences of linguistic and melodic context, and the influence of peripheral constraints in the synchronization task.
9. In response to Reviewer 2, we discussed potential reasons for the lack of an interaction between the stimulus type and the motor effector.
10. We clarified all methods questions in the manuscript.

All comments are answered in detail below.

REVIEWER COMMENTS:

Reviewer #1 (Remarks to the Author):

This study examines effects of rate in isochronous speech and tone sequences on performance in auditory-motor synchronisation and perceptual tasks. Independent experimental variables include (1) stimulus rate (2 Hz vs 4.5 Hz), (2) stimulus type (syllables vs tones), (3) motor output tested for

synchronisation (tapping vs whispering). Dependent variable (performance measure) is the ability to (1) synchronise motor output to the stimulus, or (2) detect an irregularity in the rhythm. The underlying rationale is that rate preferences might be domain-general or -specific, and this should be visible in corresponding performance effects. The authors report various observations that point to, apart from some domain-general effects, a preference of slower (2 Hz) and faster (4.5 Hz) rates for motor outputs associated with music (tapping) and speech (whispering), respectively.

R1 – Issue 1:

This is an excellent study that will be of interest to the scientific field and beyond. I do have some questions that should be seen as an effort to raise additional points of discussion rather than serious criticism.

Given the differences in dominant rates that the authors describe for music and speech (slower for the former), it would seem more natural to tap to a 2-Hz tone sequence than to a 4.5-Hz one, and vice versa for whispering. I wonder if the authors' findings are driven by typical stimulus properties that the systems are **used** to process, rather than a demonstration of actual "preferred rates" that the systems are **built** to process.

Thank you for the positive evaluation and the constructive feedback.

We agree with the reviewer that individuals are particularly used to tapping and speaking at the respective rates. It has been shown that these rates are similar across languages for speech (Ding et al., 2017) and at least simple musical rhythms are similar across cultures (Jacobi, McDermott, 2017, Current Biology), while otherwise there is a large cultural impact. Therefore, some "hardwired" neural structure has been suggested, with optimal performances being related to the endogenous brain rhythms of the auditory and motor cortices with preferred frequencies at the respective rates. Such endogenous brain rhythms have been proposed in auditory and motor cortices (Giraud et al. 2007; Keitel and Gross 2016). Whether endogenous brain rhythms evolutionary emerged as a consequence of typical stimulus properties of speech and music or whether stimulus properties emerge as a consequence of endogenous brain rhythms is a matter of ongoing scientific debate. From an evolutionary view, it has been shown that non-human primates display rhythmic facial movements such as lip-smacking, that commonly occur in the same range as human speech rhythms (MacNeilage 1998; Ghazanfar, Morrill, and Kayser 2013). Therefore, an evolutionary tuning of the speech rhythm to endogenous brain rhythms has been suggested (Ghazanfar and Poeppel 2014). However, the literature remains inconclusive in ultimately disentangling the causes and the consequences. By talking about "preferred rates" we are not making any claims with regard to this debate.

We clarified this in the introduction

L. 54: "While it is still debated whether such brain rhythms emerged from the natural properties of speech and music or whether rhythm in speech and music evolved around this functional cortical architecture³², a functional relevance has been proposed. "

R1 - Issue 2:

"Music" is defined as a random sequence of tones, and I wonder whether and how this choice might have influenced results. (a) Speculatively, what do the authors think would happen if they made these sequences "meaningful" (using, e.g., melodies), or used tasks centred on music or speech perception (i.e. on their interpretation rather than perception)? (b) What do the authors consider the defining feature in the tones that has caused the domain-specific preference for slower rates – is it enough to present sounds with a clearly defined pitch (driven by a single sound frequency) to produce such music-

specific effects? (c) What type of stimuli would the authors use to contrast general auditory (rather than music-specific) effects with speech-specific ones?

(a) Making our stimulus sequences “meaningful” by adding higher-level context (e.g., presenting sentences or melodies) or by using “comprehension tasks” might result in top-down modulations as well as a different focus of attention possibly modulating the effects. On the other hand, beat perception has been suggested to be a robust perceptual phenomenon that occurs even in the absence of acoustic beat cues (Large, Herrera, and Velasco 2015). Similarly, neural tracking of syllables has been observed even when rhythmicity at this time scale is removed from the acoustics (Zoefel and VanRullen 2016). In summary, although our results might be modulated by task and context changes, we would expect the processing at these time scales to be robust to some extent.

(b) There is evidence suggesting that stimulus characteristics can trigger speech and music-specific processing. A recent study suggested that the classification of amplitude modulated noise as speech or music is related to the rate of amplitude modulation, with slow rates around the beat level triggering music perception and faster rates speech perception (preprint: Chang et al. 2022). Similarly, Albouy et al. (preprint: 2023) compared speech and song across cultures and found that stimuli can be reliably classified as speech or song based on their spectro-temporal features (with temporal modulations at the syllabic rate being relevant for speech, while song shows slower temporal modulations and spectral modulations are more characteristic). Others suggested that in the Deutsch-Deutsch speech-to-song illusion, the perception of song emerges when speech contains a more periodic rhythmic structure compared to that inherent in speech (Tierney, Patel, and Breen 2018; Vanden Bosch der Nederlanden et al. 2022). Speech specific neural processing might also be driven by other acoustic characteristics (Overath et al. 2015). In our study, participants listened to syllable sequences or piano tone sequences. Although, this is simplified “speech” or “music” that is lacking higher-order context, it can be expected that when participants perceive syllables, they recognize the stimulus as speech and activate the respective processing. Similarly, perceiving piano tone sequences is expected to activate music-related processing. Thus, although certain characteristics of the acoustic stimuli might be enough to trigger specific processes, syllables and piano tones should activate the respective processes in any case.

(c) Chang et al. (preprint: 2022) found that faster amplitude modulations $\sim 4.5\text{Hz}$ were robustly related to the categorization of sound as speech, the effect was less consistent for music and depended more on the listeners experience with music. Thus, it is possible that no strong distinction between music-specific and general auditory processing is drawn. In order to compare these effects, future studies should use experimental conditions comparing natural sounds contrasted with music and speech signals.

We clarified these points in the discussion.

I. 361: “Our study has a limited scope in stimulus material and motor effector choice (i.e., syllable and piano tone sequences instead of natural speech and music and whispering and finger-tapping instead of natural speech and music production). However, the benefit is that our speech and music conditions are well matched acoustically, and we show that our results are not merely caused by differences in the acoustics. We refrained from using more complex stimulus material in order to enable a close matching of the syllable and piano tone sequences. However, investigating how additional contextual information affects optimal processing rates in perception and production requires future research.”

I. 375: “Our findings do not aim to speak towards the minimal acoustic features that are required to elicit speech or music-specific processing, which have been researched elsewhere^{96,97,105}, (preprint:^{76,98}).”

R1 – Issue 3:

Introduction: “Speech and music production typically employ distinct articulator systems”. But mouth and vocal cords are enough to make music, and hands and arms are also employed during human communication. Do the authors predict different preferred rates for vocal music?

We agree that it is a simplification to use the mouth and vocal cord (whispering) and particularly the fingers (finger-tapping) as motor effector systems used in speech and music. Our choice is based on the following: First, these systems are typically researched in the speech and music fields. Especially results from finger-tapping tasks have shaped our views about auditory-motor synchronization in general and rhythm processing in music in specific (using rhythmic click tones, pure tones sequences, or music, for review, see: Repp, 2005; Repp & Su, 2013). Finger-tapping has been widely used to investigate spontaneous production rates (McAuley et al., 2006; Repp & Su, 2013; Zalta et al., 2020; Tranchant, Scholler, Palmer, 2022) and synchronization (Scheurich, Zamm, Palmer, 2018; Tranchant, Scholler, Palmer, 2022; Roman et al., 2023). To our knowledge, auditory-motor synchronization of speech production has only more recently been investigated (e.g., Assaneo et al., 2019), and has not been directly compared to finger-tapping (however, see a very recent publication this June in Communications Biology that compared whispering and clapping: Mares et al., 2023). Second, importantly, although other motor effectors might be involved in music production, the finger/hand seems an adequate approximation for motor effectors associated with piano playing, that is, the sound we used.

Vocal music provides an interesting case, as similarly as in speech the mouth and vocal cord are used, while the musculature, breathing, and motor control mechanisms differ (Sundberg, 1989). A recent large-scale cross-cultural comparison study of speech and song (preprint: Albouy et al., 2023) finds that spectro-temporal features of speech and song are clearly distinct, with song showing similar dominant time scales as music in general (< 2 HZ) and a similar focus on pitch information (with longer syllable durations allowing for more stable pitch production and tonal encoding). Thus, vocal music might still activate distinct rhythmic motor timing compared to speech, as the motor effectors are employed in a distinct fashion.

We clarified this in the manuscript.

I. 100: “Speech is produced by the mouth (lips, tongue, jaw) and the vocal cords. Other motor effectors such as the hands and arms can additionally support non-verbal aspects of speech production. Non-vocal music production commonly relies on the hands and arms (or sometimes the feet). For singing, the mouth and the vocal cords are used, though in a different manner when compared with speech (preprint: ^{76,77}). “

I. 333: “Vocal music may provide an interesting case for future research. Speech and song overlap with regard to their motor effectors, while song shows acoustic characteristics similar to that of non-vocal music ^{96,97}(preprint: ⁹⁸). This has been related to a different engagement of the motor effectors. Therefore, we expect singing synchronization to recruit rhythmic motor timing associated with the music domain that is optimal at slow time scales.”

R1 - Issue 4:

“Whispering synchronization was less disadvantaged compared with tapping at fast rates, which is consistent with research indicating that the speech system is optimized at faster rates than the music system.” Could such a result not simply stem from differences in properties of muscles required (e.g.,

we cannot move fingers as quickly as we can move the jaw)? Unless the authors consider these to be part of specific “speech and music systems” (which might be good to state explicitly)?

In our study, additionally to the synchronization task, we used a perception task that did not involve overt production. As we confirm the optimal time scales for syllable and piano tone sequence processing in this task, our findings seem to reflect higher-level motor timing and not peripheral muscle constraints. We now clarified this in the discussion:

I. 303: “Alternatively, the observed effects could result from peripheral constraints such as the ability to move our jaw faster than our fingers. This, however, cannot account for our findings in the perception task in which no overt production was required. We therefore suggest that the findings reflect the recruitment of higher-level rhythmic motor timing in speech and music rather than, or in addition to, differences in peripheral muscle movements. “

R1 - Issue 5:

“Faster beta brain rhythms around 20 Hz and slower delta brain rhythms around 2 Hz [...]”. I found this sentence a bit confusing, as 2 Hz and 4.5 Hz are used for stimulus presentation in the experiment, and 20 Hz does not appear to be relevant.

We agree that the beta brain rhythm is not relevant for the study’s conception and therefore removed it from the introduction.

R1 – Issue 6:

I leave it to the authors, but the recent paper by Zuk et al (2021, Plos Comp Biol) might be an interesting addition to the manuscript.

Thank you for pointing out the reference.

The authors interpret their finding of the superior performance of domain-general models when compared with stimulus-specific model (using a TRF envelope reconstruction approach) for speech and music at faster frequencies above 1 Hz as evidence for general mechanisms involved in speech and music processing. This finding is interesting and might partly fit our findings of a shared component for synchronization at the faster rates although they use a much broader frequency range and different methodology.

We now cite the reference when we discuss our findings.

I. 330: “Furthermore, a common mechanism for the neural tracking of speech and music at faster rates has been suggested (with other findings of this study, however, being in contrast to ours and direct comparisons being hindered because of broader frequency ranges and other methodological differences) ⁷⁵. “

The other findings of the study are seemingly contradictory to the findings we based our hypothesis on: The authors report no peak at beat related time scales in their music stimuli and they find no peak for the tracking. Human beat perception has been shown to be optimal around 1-2 Hz (London 2004; Cannon and Patel 2021) and this has been reported as the time scale containing the dominant acoustic energy in the modulation spectrum if a larger corpus of music from diverse styles is considered (~0.5-3 Hz; Ding et al., 2017; see also: Zhang et al., 2023), with some variation. In contrast to Zuk et al. (2021), others have reported neural tracking of music at the beat rate (e.g., Zhao & Kuhl, 2020; Morillon & Baillet, 2017). In Zuk et al. (2021), the EEG was recorded during passive listening. Therefore, neural

tracking might be affected by several processes that are hard to disentangle. For example, the increased tracking of speech compared to music at all frequencies might be related to attention effects or differences in predictability. The increased tracking of speech at frequencies < 1 Hz might reflect higher-level phrasal processing. Therefore, we refrain from making further direct comparisons with our findings.

We now clarified in the introduction that speech and music contain rhythmic structure at several time scales and cite the reference here.

I. 92: "It should be noted that besides these dominant rhythmic modulations, speech and music also contain several hierarchical levels of information with rhythmic modulations at different time scales⁶⁷. For example, speech contains rhythmicity beyond the syllable level^{20,39,42,68} at the phrasal level at around 1 to 2 Hz^{33,34,69-74}. Music contains rhythmic fluctuations beyond the beat rate at faster single note rates or slower phrasal rates^{18,75}."

Reviewer #2 (Remarks to the Author):

This study investigated rate-dependent differences between speech and music using a perception and a synchronization task. The main finding of the perception task was that tone perception was optimal at slower rates (~2 Hz), syllable perception was optimal at faster rates (~4.5 Hz). In the synchronization task, the authors found a higher PLV between auditory stimuli and both articulators at slower rates than the faster rate, and a higher PLV with the tone stimuli than the syllable stimuli. Further, significant interaction effects showed that PLV was higher when tapping than whispering, as well as synchronizing to tones than the syllables, only at slower rates rather than the faster rates. The authors concluded there is a domain-specific effect of articulator and stimuli at slower rates while a domain-general effect no matter of the articulator or stimuli at the faster rates. The study was pre-registered. Some analyses deviated from the original plan but were clearly explained in the supplementary material.

This work is interesting and it is worthwhile to examine the sources of rate-dependent effects on speech and music. It's an important question, thus the magnitude of comments below. At present there are a few major conceptual and analysis issues that must first be addressed that prevent it from reaching this journal's required bar of influencing thinking in the field. The most serious reservation is that comparing single-speaker speech modulation to ensemble music modulation is totally incommensurate. The 2Hz modulation of music is not only (and perhaps not even primarily) to do with **(a)** articulator neuromuscular dynamics (especially a single finger) but is a constraint of it being an ensemble activity linked by an internalized sense of beat. This seems like a major showstopper. Less fatal, but in need of some more justification is that calling the finger the music articulator and voice (uniquely) the speech articulator seems to overly trivialize the reality of music. There are a number of additional, smaller needed clarifications, most of which all seem readily addressable.

Thank you for the positive evaluation and the criticism that helped us to thoroughly revise the manuscript.

General comments

R2 – Issue 1:

(b) The hypotheses could be more clearly stated. What are the hypotheses among tempo, stimuli, and articulator? Do the authors assume that different articulators are linked to distinct motor processing circuits, which have different rate sensitivities? (c) If so, do they further assume multiple domain-specific time processing circuits? Please spell out the fundamental mechanistic hypotheses of this study.

(a) Our claims are not about neuromuscular dynamics but about rhythmic motor timing mechanisms in the brain and their effects on behavior. This has been clarified throughout the manuscript.

(b) We behaviorally test the hypothesis that different motor effectors are linked to distinct rhythmic motor timing circuits with specific rate sensitivities. The hypothesis is based, on the one hand, on findings of different dominant rates in the produced speech and music acoustics (Ding et al., 2017; Poeppel & Assaneo, 2020) and on the other hand, on behavioral evidence for different spontaneous production rates in speech (Pellegrino, 2011; Lubinus, 2023) and music (Zalta et al., 2020; Scheurich et al., 2018; Tranchant et al., 2022). We clarified this in the manuscript:

l. 77: “Analyses of large corpora of produced speech and music signals revealed that for diverse types of music played on various instruments, slow acoustic amplitude modulations around 1-2 Hz are dominant^{13,15}. [...] In contrast, speech shows faster dominant amplitude modulations at the syllabic rate around 4 to 8 Hz across languages^{13,15,62}.”

l. 109: “Differences related to motor effectors have also been reported in the context of spontaneous production rates. Rhythmic motor timing in music has been traditionally researched in finger-tapping paradigms⁷⁹⁻⁸³. Spontaneous finger-tapping rates have been observed around 2 Hz^{60,81,82,84-86}, with optimal synchronization of finger-tapping to the beat at these rates^{81,82,87}. The repetition of piano melodies by trained pianists has revealed similar spontaneous rates around 2 Hz⁸⁸, which were correlated with the individual spontaneous finger-tapping rates. Fewer studies investigated spontaneous syllable production rates and found optimal rates around 4 to 8 Hz in natural speech production^{27,62}. Other methods require individuals to repeatedly whisper a single syllable, and confirmed spontaneous rates around 4 to 5 Hz⁸⁹. “

Second, this hypothesis is based on neural research. Typically, rhythmic motor timing circuitry have been investigated separately for speech and music. Although there is some overlap in the neural circuitry proposed to be relevant for rhythmic motor timing in speech and music, overlapping brain areas might be recruited differentially for music and speech processing, as suggested recently (preprint: Rietmolen et al., 2022). We clarified this in the introduction (ll. 69, see below).

(c) Previously, one (possibly domain general) rhythmic motor timing mechanism has been proposed for slow rates around 2 Hz related to beat perception (Cannon & Patel, 2021), which might allow to derive timing at other rates. While others have suggested multiple parallel “sampling” mechanisms for speech processing (Poeppel, 2003). We focus on the dominant rhythmic time scales and had no clear hypothesis about whether multiple domain-specific rhythmic timing circuits exist or timing at other time scales is derived from a dominant mechanism.

We clarified this in the abstract, in the introduction and in the hypotheses section:

l. 33: “Our data suggests partially independent rhythmic timing mechanisms for speech and music, possibly related to a differential recruitment of cortical motor circuitry.”

L. 66: “The supplementary motor area (SMA) and the basal ganglia have been suggested to function as a “pacemaker” during speech perception^{51,52} and particularly during beat perception and anticipation in music^{17,53,54}. Particularly, slow delta brain rhythms around 2 Hz observed in the SMA seem to be involved in temporal predictions provided by the motor system^{26,29,31,55}. This time scale corresponds to the time scale of beats in music^{17,31,56-58}, while its role in speech processing is not fully understood. However, delta brain rhythms around 1-2 Hz have been suggested to support domain-general rhythmic motor timing^{17,31}. “

I. 90: “Accordingly, on a neural level, overlapping brain areas recruited for speech and music processing^{3,4,6} have been suggested to show frequency-specific selectivity for speech and music (preprint: ⁶⁶).”

I. 131: “In a behavioral paradigm, we tackle the question of domain-specific mechanisms by investigating whether the optimal time scales in the speech and music domain differ and depend on the motor effector involved in their production. The optimal rate was defined as the stimulus presentation rate with highest performance. In a perception-production synchronization task as well as an auditory perception task, we used speech (syllable sequences) and music stimuli (piano tone sequences) and two different motor effector systems (whispering and finger-tapping). All tasks were performed at slow rates around 2 Hz (1.92 – 2.08 Hz) and fast rates around 4.5 Hz (4.3 – 4.7 Hz). We hypothesized that specific motor effectors recruit distinct cortical rhythmic motor timing circuitry with distinct optimal processing rates that constrain the auditory-motor coupling. More specifically, we predicted that the involvement of motor effectors associated with speech is related to higher synchronization performance at fast rates around 4.5 Hz, while motor effectors associated with music show highest synchronization performance at slower rates around 2 Hz. Assuming that the corresponding motor systems are activated even without overt motor behavior in the auditory perception task⁴³⁻⁴⁶, we hypothesized that the performance in the perception task should mirror the results from the synchronization task, with higher and lower rates enhancing speech and music processing, respectively. Furthermore, synchronization was expected to predict perception performance at the corresponding time scale. Alternatively, we hypothesized that rhythmic timing processes facilitated by the motor system might generally be optimal at slower time scales, which has been suggested in previous work^{17,31}. This would result in higher performance at slow time scales across domains. “

R2 – Issue 2:

The link between the articulators and certain stimuli, or domains, could stand to be further explained. The authors designed this study by presenting finger tapping as a “music articulator” and whispering as a “speech articulator”. What is exactly is meant by a whispering articulator, since speech requires the coordination of multi-articulators such as mouth, lip, tongue, vocal cord, etc. Was whispering chosen in order to focus on a single vocal articulator? It is not clear. It would help if the authors more clearly explained the reasoning of selecting these two articulators and consider measuring the average rate and range of whispering and tapping rates to present a kind of ground truth of production abilities.

Similar to speaking, whispering uses the mouth and the vocal cords for speech production, however, vibration of the vocal cords is reduced compared to normal speaking. We did not aim at selectively testing only certain motor effectors involved in speech production. We used whispering because it has been previously used in the SSS test. It was used in the original SSS test in order to avoid feedback from the auditory system while synchronizing speech production to perceived speech. We clarified this in the discussion by mentioning that using whispering instead of speaking provides a potential limitation.

L. 368: “Additionally, a potential limitation of our work is the use of whispering instead of natural speaking in the synchronization task. The rationale behind this decision – following the protocols of the SSS test^{90,92} – was that auditory feedback from one’s own speech production was minimized by the low tone of voice. As whispering involves the mouth and vocal cords in a very similar manner as speaking (while the vocal cords are not vibrating), we would not expect differences in motor effector associated rhythmic timing⁹⁰. Findings from the perception task, in which spoken syllables (no whispering) were used, are in line with this assumption. “

We agree that it is a simplification to use the mouth and vocal cord (whispering) and particularly the fingers (finger-tapping) as motor effector systems used in speech and music. Our choice is based on the following: First, these systems are typically researched in the speech and music field. Especially results from finger-tapping tasks have shaped our views about auditory-motor synchronization in general and rhythm processing in music in specific (using rhythmic click tones, pure tones sequences, or music, for reviews, see: Repp, 2005; Repp & Su, 2013). Finger-tapping has been widely used to investigate spontaneous production rates (McAuley et al., 2006; Repp & Su, 2013; Zalta et al., 2020; Tranchant, Scholler, Palmer, 2022) and synchronization (Scheurich, Zamm, Palmer, 2018; Tranchant, Scholler, Palmer, 2022; Roman et al., 2023). To our knowledge, auditory-motor synchronization of speech production has only been investigated more recently (e.g., Assaneo et al., 2019), and has not been directly compared to finger-tapping (however, see a very recent publication this June in Communications Biology that compared whispering and clapping: Mares et al., 2023). Second, importantly, although other motor effectors might be involved in music production, the finger/hand seems an adequate approximation for motor effectors associated with piano playing, that is, the sound we used.

We clarified this in the manuscript.

I. 100: “Speech is produced by the mouth (lips, tongue, jaw) and the vocal cords. Other motor effectors such as the hands and arms can additionally support non-verbal aspects of speech production. Non-vocal music production commonly relies on the hands and arms (or sometimes the feet). For singing, the mouth and the vocal cords are used, though in a different manner when compared with speech (preprint:^{76,77}). “

Several previous studies included measurements of spontaneous tapping and speaking rates. Spontaneous tapping rates have been consistently observed around 2 Hz (Moelants 2002; McAuley et al. 2006; Zalta, Petkoski, and Morillon 2020; Kaya and Henry 2022). Additionally, when participants were asked to tap at their slowest and fastest rates, those ranging from 0.2 up to more than 6 Hz were produced (Zalta, Petkoski, and Morillon 2020; Kaya and Henry 2022). In speech, for reading, syllable rates have been shown to range from 5 to 8 Hz across languages (Pellegrino, Coupé, and Marsico 2011). In spontaneous speech production, a recent study has revealed syllable rates ranging from 3.36 to 5.38 (Lubinus et al. 2023). Crucially, a previous study investigated the spontaneous rates when whispering a single syllable and revealed similar mean rates ranging from 3.5 to 5 Hz (Assaneo et al. 2021). These spontaneous rates correspond to the rates we chose as fast and slow rates in our study. Based on the previous research, we assume that these rates are appropriately capturing the dominant spontaneous rates in both domains.

Additionally, we added the above-mentioned references to the introduction to back up the spontaneous rates in whispering and tapping.

I. 111: “Spontaneous finger-tapping rates have been observed around 2 Hz^{60,81,82,84-86}, with optimal synchronization of finger-tapping to the beat at these rates^{81,82,87}. The repetition of piano melodies by trained pianists has revealed similar spontaneous rates around 2 Hz⁸⁸, which were correlated with the

individual spontaneous finger-tapping rates. Fewer studies investigated spontaneous syllable production rates and found optimal rates around 4 to 8 Hz in natural speech production^{27,62}. Other methods require individuals to repeatedly whisper a single syllable, and confirmed spontaneous rates around 4 to 5 Hz⁸⁹.”

R2 – Issue 3:

Unfortunately, the entire finger/music voice/speech dichotomy is less than clear! Many musical behaviors require both the mouth and fingers or the mouth and no fingers, e.g. wind instruments and singing. There is very little music that involves a single digit.

We clarified our reasoning for the selection of motor effectors in our response to R2 – Issue 2.

R2 – Issue 4:

(a) Most fundamentally, the comparison of differences in peak modulation frequency for speech and music seems like a red herring and an apples to oranges comparison. The 'faster' speech modulation result is for a *_single_* speaker, while the 'slower' music modulation rate is for *_musical ensembles_* with multiple instruments. These are not commensurate at all! A proper comparison, if there even is one, might be to look at the modulation rate of e.g. musical solos on instruments ideally played only with a single articulator. (b) (In reality, instruments played with multiple digits can have a much higher frequency of AM relating to individual notes, not the 'beat rate' modulation that emerges in ensemble music.) This neglects secondary aspects of musical articulation such as accents and loudness, which occur at rates slower than the main articulation—that is something else entirely than simple SMS). As I think may have been mentioned, there is a key difference between tactus rate and beat rate. Not the same thing at all.

(a) This seems a misunderstanding. In our work, we do not include ensemble music. We compare single speaker syllable sequences to single instrument piano tone sequences. Crucially, in previous work the dominant 2Hz modulation of music has been shown for several types of music including single-instruments music, which showed a similar temporal modulation spectrum (Ding et al. 2017).

(b) For music processing the beat rate has been claimed to be special as rhythmic motor timing seems to optimally operate at this rate, and timing at other rates might be reconstructed based on the beat timing (Cannon and Patel 2021). Of course, music production includes movements at faster rates (e.g., single notes), however, most energy in the acoustic modulation spectrum has been reported at the beat rate (also suggesting that this might be the most “periodic” occurrence; Ding et al., 2017). In our study, we investigate whether such rhythmic timing mechanisms are recruited during the synchronization and perception of syllables and piano tones.

In conclusion, although more natural stimuli could be used, we believe our stimulus material is adequate to investigate our claims. We clarified this in the introduction.

I. 81: Although the beat might be crucial for interpersonal coordination in musical ensembles, the dominant temporal modulations at slower rates are equally observed in ensemble and single instrument music¹³.”

Additionally, we now clarified in the introduction that speech and music contain rhythmic structure at several time scales and cite the reference here.

I. 92: “It should be noted that besides these dominant rhythmic modulations, speech and music also contain several hierarchical levels of information with rhythmic modulations at different time scales⁶⁷. For example, speech contains rhythmicity beyond the syllable level^{20,39,42,68} at the phrasal level at around 1 to 2 Hz^{33,34,69-74}. Music contains rhythmic fluctuations beyond the beat rate at faster single note rates or slower phrasal rates^{18,75}.”

R2 – Issue 5:

Possible solutions: There are probably some bass lines that are played with a single finger around the beat rate. Alternatively, for the speech side a better comparison would be to look at the modulation of many people chanting together. I suspect that will slow to be close to music—However, the fundamental underlying and basic point is that ~2Hz modulation is likely a constraint of _multi-person synchronization_ (mediated through a shared beat) and not an articulatory constraint!

Again, we apologize for the misunderstandings about our stimuli, which are matched with regard to coming from a single piano player/single speaker. The reviewer suggests that the signal for a “many people chanting together” scenario would look close to music. Interestingly, it has been recently shown that single voice song looks closer to music in terms of the dominance of slow temporal modulations (preprint: Albouy et al., 2023). Thus, this effect does not seem to require multi-person synchronization. Instead, Albouy et al. (2023) suggest that the effects may “result from differences in how human vocal musculature is used for speaking or singing”. The slowing of speech production at the syllabic rate during singing has been suggested to give more room for a stable pitch leading to better encoding of tonal relationships important for music (see also: Mantell & Pfordresher, 2013). We agree that the role of the beat in ensemble coordination is an important topic, however, as we clarify in detail in R2 Issue 4 and R2 Issue 1, we matched our conditions with that regard and, beyond this, see no problem for our study.

R2 – Issue 6:

Based on the preceding items, which made the same point in multiple ways, it becomes hard to see how the logic of this study makes sense, though am open for it to be further clarified and justified that it makes sense.

We did a thorough revision and rewriting of the introduction and discussion and sincerely hope that our reasoning is now clear and convincing (see our detailed responses above).

R2 – Issue 7:

The Introduction thoroughly reviewed the neural evidence regarding speech and music but did not mention much on the behavioral evidence. Given that this is a behavioral study, please present more behavioral evidence background.

We adjusted the introduction accordingly; examples are:

I. 85: “Furthermore, different rhythmic characteristics of speech and music were not only observed in the produced signals but are also reflected in the perceptual performance. For example, beat deviance detection in pure tone sequences has been shown to be maximal for beat rates of about 1.4 Hz⁶⁰. In contrast, speech comprehension performance has been suggested to be highest for syllable rates in the theta range (~4.5 Hz) and drop at faster rates around 9 Hz^{63,64} (or at even higher rates^{27,65}). “

I. 111: “Spontaneous finger-tapping rates have been observed around 2 Hz^{60,81,82,84-86}, with optimal synchronization of finger-tapping to the beat at these rates^{81,82,87}. The repetition of piano melodies by trained pianists has revealed similar spontaneous rates around 2 Hz⁸⁸, which were correlated with the individual spontaneous finger-tapping rates. Fewer studies investigated spontaneous syllable production rates and found optimal rates around 4 to 8 Hz in natural speech production^{27,62}. Other methods require individuals to repeatedly whisper a single syllable, and confirmed spontaneous rates around 4 to 5 Hz⁸⁹. “

R2 – Issue 8:

There is one analysis flaw that affects the main conclusions: The critical point that acoustic features of the stimuli had a very large influence on the results was underplayed and placed in the supplementary materials. We think that the acoustic features are essential to address in the main text. Specifically, Line 217 described the results as not being driven by the envelope characteristics, but comparing Table 2 to Supplementary Table 1 this is clearly not the case: All findings except for the Tempo effect were lessened or erased by controlling for the envelope characteristics. Based on the results, the envelope characteristics does not seem like a confound. Instead, it is likely a very important factor driving the main PLV results and therefore should be the main analysis. Onset profile effects are mentioned at line 167 in text.

The envelope characteristics are an important factor for the processing of speech and music. However, there seems to be a misunderstanding: Including them into our model had weakened some of the effects. All effects relevant for our claims, however, were still significant (i.e., the main effect of rate and the interaction of rate and motor effector).

Following the reviewers’ recommendation, we now included the analysis in the main manuscript and adjusted the procedure to our main analysis by using a step-wise regression. The step-wise regression maximizes the model fit while discarding non-significant predictors. In the step-wise regression (as in our previous control analysis), only the width of the peaks of the motor envelopes explained a significant share of variance in synchronization performance. Interactions involving the stimulus type did not explain a significant share of variance and were therefore not included in the model. Consequently, only the fixed effects of stimulus type, motor effector, rate, and motor envelope peak width, as well as the interaction between rate and motor effector were included in the model. Additionally, as a result of the reviewers’ suggestion of adjusting the bandwidths used for filtering the auditory envelopes (see R2 – Detailed issue 35), the model converged when we added a random slope for the rate, which was not possible before. This new model compared to our previous control model (that included all envelope characteristics and interactions between rate, stimulus, and motor effector) provided a better fit to the data based on the AIC (AIC_{old} = -864.36, AIC_{new} = -962.44). The methods, results and table 2 were adjusted accordingly.

We adjusted the results section as follows and we adjusted Table 2 and the methods section accordingly. Additionally, we adjusted Figure 2 based on the updated results. The Figure now shows the interaction between rate and motor effector along with the individual model estimates and the random slopes.

I. 171: “As we were interested in endogenous rhythmic timing mechanisms that are not reflecting processing advantages related to acoustic signal differences, we controlled for acoustic envelope characteristics in the model. We therefore added characteristics of the envelope of the speech and music signal (acoustic envelope) and of the envelope of the recorded whispering and tapping signal (motor envelope) as predictors. Characteristics of the acoustic envelope provide crucial landmarks for

the neural tracking of speech and music^{18,20,95}, and may contribute to the perception of stimuli as speech or music^{96,97} (preprint: ^{76,98}). The recorded whispering and tapping signals (motor envelope) might differ and confound the synchronization measure. The step-wise regression procedure revealed that only the motor envelope characteristics significantly improved the model fit and thus explained variance in synchronization performance. The acoustic envelope characteristics were therefore not included in the model. Descriptively, the motor envelope peak width was larger for tapping compared to whispering and for slow rates compared to fast rates. A larger envelope peak width was related to higher synchronization performance ($p < .001$), which indicates an improved synchronization to the accelerating rhythmic structure. “

Accordingly, table 3 for the perception task and the methods and results for the perception task were revised.

I. 230: “The results additionally revealed a significant effect of the width of the peaks of the acoustic stimulus envelope on perception performance. Thus, characteristics of the stimulus envelope influenced perception performance, with a smaller peak width being related to higher performance. Descriptively, the peak widths were larger for the syllable sequences than for the piano tone sequences, which could be expected given the acoustic characteristics of piano tones compared to syllables.”

Detailed comment

1. Line 24: This sentence is a bit confusing. How can music and speech rhythmic structure have emerged from the auditory or motor system?

The sentence and the abstract have been revised to improve clarity.

I. 24: “Speech and music might involve specific cognitive rhythmic timing mechanisms related to differences in the dominant rhythmic structure.

2. Line 29: Could spell out the slow and fast rate in the Abstract for clarity.

We included the slow and fast rates in the abstract:

I. 26: “A perception and a synchronization task involving syllable sequences and piano tone sequences as well as motor effectors typically associated with speech (whispering task) and music (finger-tapping task) were tested at slow (~2 Hz) and fast rates (~4.5 Hz).”

3. Line 34: Not sure how the synchronization strength is defined.

We clarified this in the methods section:

I. 501: “In the synchronization task, the synchronization strength between the envelope of the acoustic signal and the envelope of the motor output was measured using the phase-locking value (PLV) between both signals (with 1 denoting strong synchronization and 0 no synchronization). “

Note that a normalization was applied to the PLVs (I. 523)

4. Line 58: Please clarify the “rhythmic characteristics”

This has been clarified.

I. 76: “In spite of their significant overlap, the produced speech and music signal show crucial differences in rhythmic characteristics. Analyses of large corpora of produced speech and music signals revealed that for diverse types of music played on various instruments, slow acoustic amplitude modulations around 1-2 Hz are dominant^{13,15}. Interestingly, this rate corresponds to the preferred rate of human beat perception^{59,60}, and beat perception has no equivalent in the speech domain⁶¹. Although the beat might be crucial for interpersonal coordination in musical ensembles, the dominant temporal modulations at slower rates are equally observed in ensemble and single instrument music¹³. In contrast, speech shows faster dominant amplitude modulations at the syllabic rate around 4 to 8 Hz across languages^{13,15,62}.”

5. Line 59: Please discuss more about what specific areas in the motor system are referred to here

We now discuss potential motor regions involved in rhythmic timing.

I. 66: “The supplementary motor area (SMA) and the basal ganglia have been suggested to function as a “pacemaker” during speech perception^{51,52} and particularly during beat perception and anticipation in music^{17,53,54}. Particularly, slow delta brain rhythms around 2 Hz observed in the SMA seem to be involved in temporal predictions provided by the motor system^{26,29,31,55}. This time scale corresponds to the time scale of beats in music^{17,31,56-58}, while its role in speech processing is not fully understood. However, delta brain rhythms around 1-2 Hz have been suggested to support domain-general rhythmic motor timing^{17,31}. “

6. Line 71: Music production is not necessarily employed by arms and hands. Actually, vocal cord has been used in many forms of music production.

In R2 issue 2, we discuss our reasoning for choosing these motor effectors in detail, and we clarified and more carefully worded this throughout the manuscript (see example below).

I. 100: “Speech is produced by the mouth (lips, tongue, jaw) and the vocal cords. Other motor effectors such as the hands and arms can additionally support non-verbal aspects of speech production. Non-vocal music production commonly relies on the hands and arms (or sometimes the feet). For singing, the mouth and the vocal cords are used, though in a different manner when compared with speech (preprint:^{76,77}). “

I. 110: “Rhythmic motor timing in music has been traditionally researched in finger-tapping paradigms⁷⁹⁻⁸³. “

Vocal music is an interesting case, we discuss this in detail in R1 issue 3 and added it to the discussion:

I. 333: “Vocal music may provide an interesting case for future research. Speech and song overlap with regard to their motor effectors, while song shows acoustic characteristics similar to that of non-vocal music^{96,97}(preprint:⁹⁸). This has been related to a different engagement of the motor effectors. Therefore, we expect singing synchronization to recruit rhythmic motor timing associated with the music domain that is optimal at slow time scales.”

7. Figure 1: Suggest adding “temporal” in front of the deviation

This was revised in the figure. Note that the figure was additionally revised for clarity.

8. Table 2: There is no significant interaction effect between Articulator and Stimulus on PLV (as well as

Supp. Table 1), is this expected by the authors? Isn't this finding against the hypothesis that certain articulator is preferred in certain domain (i.e. music vs. speech here)?

We expected that both the stimulus type and the associated motor effector trigger the respective rhythmic timing mechanism. Thus, we would have expected an interaction effect between motor effector and stimulus on PLV. Although, in the synchronization task we did not observe this interaction, we did observe it in the perception task. Here, we found an interaction between stimulus type and rate, suggesting that the stimuli did engage the respective motor effectors. We now discuss possible explanations:

I. 309: "Additionally, the results did not reveal any interaction between stimulus type and the motor effector or the rate, which we expected based on the close association of stimulus types and motor effectors. Interestingly, we show the expected interaction of stimulus type and rate in the perception task, indicating that the syllable and piano tone sequences did indeed activate the respective rhythmic timing mechanisms.-A possibility is that the fixed effect of the stimulus type dominated in the synchronization task, as synchronization performance was overall higher for piano tones compared to syllables across conditions. In the perception task, we had controlled for an overall effect of stimulus type by matching the task difficulty across conditions."

9. Line 132-3: This is a big assumption, please justify.

We justified the assumption that the corresponding motor systems are activated even without overt motor involvement in the introduction and we added additional references

L. 61: "The motor system involvement in rhythmic timing is in accordance with a - for obvious biological reasons - tight coupling of sensory and motor systems in the speech and in the music domain. The motor regions involved in production have been shown to be activated solely by listening to speech^{43,44} and music⁴⁵⁻⁴⁷. Temporal motor prediction has been shown to support speech processing in demanding listening conditions^{48,49}, as well as music processing^{17,31,50}. "

L. 143: "Assuming that the corresponding motor systems are activated even without overt motor behavior in the auditory perception task⁴³⁻⁴⁶[...]"

10. Line 139: were training/experience effects taken into account? People naturally vary a lot in SMS ability depending on if and which instrument they may play.

Participant's self-reported musical sophistication was accessed using the Goldsmith Musical Sophistication Inventory (Gold-MSI). In the supplementary material we had reported the correlation of the PCA components with the Gold-MSI score and the perception model with the Gold-MSI score as control variable. We now added a model of the synchronization data that includes the Gold-MSI score as control variable, as well as its potential interactions with rate, effector, and stimulus.

Although musical sophistication had an effect on synchronization performance and it interacts with the rate and the effector, all other effects remain constant. These findings are in line with our previous studies (Kern et al., 2021; Rimmele et al., 2022). Additionally, in Rimmele et al. (2022) we analyzed the relation of auditory-motor synchronization (SSS test) to musical sophistication in more detail. We did not find an effect of the type of instrument played on the synchronization ability. However, this was possibly due to not investigating professional musicians, and small sample sizes in some instrument groups, e.g., few participants playing percussion instruments.

See also discussion I. 355: “Interestingly, we found that only the fast synchronization PCA component – that generalized across motor effectors – was highly correlated with musical sophistication (supplementary analysis 2).”

11. Line 152: Why were two different models (LMM vs GLMM used)? On my understanding, it would only make sense if a different link function was needed for GLMM. Relatedly, the text here mentions only GLMM (presumably linear link) but the supplementary materials discuss it as a logistic model.

We use the LMM for the synchronization task, since the dependent variable is continuous. In the perception task, the dependent variable (i.e., accuracy) is binary. Therefore, we use a logistic mixed effects model, as a subtype of generalized mixed effects models. We now specified the subtype.

I. 161: “We calculated a generalized linear mixed effects model (GLMM) using a logistic link function to predict the performance accuracy in each trial based on the presentation rate and the stimulus type (piano tones versus syllables).”

12. Line 155/Figure 1. Key methodological question: how were sounds sped up/slowed down? Were syllables stretched or only the onset times adjusted? The former would be problematic since it would alter the temporal sharpness of the onset. Please clarify.

The syllable sequences were synthesized by the MBROLA speech synthesizer at different rates. MBROLA is based on the concatenation of diphones, taking as input a list of phonemes together with prosodic information (i.e., duration). Rates are controlled by indicating syllable durations. Similarly, the piano tone sequences were synthesized using MIDI files with different tempi, which in turn controlled the tone durations. No stretching or adjusting of onset times was required. We clarified this in the methods.

I. 424: “All stimuli were synthesized at their respective rate, based on the syllable and tone duration information provided to the synthesizer.”

13. Line 158: to be fair, these are weak interactions.

This has been clarified in the discussion.

I. 307: “Despite their high significance, it should be noted that the magnitude of the effects in the synchronization task was rather small.”

14. Line 162: The Results are very well written and easily to follow along with Table 2!

Thank you.

15. Table 2: However, confused as Articulator x Stimulus would seem to be the key factor in supporting the main hypothesis of the paper, yet it is not significant.

We agree that this interaction could be expected from the hypothesis. We named potential reasons for the lack of the effect in our response to R2 – Detailed comment 8. Additionally, we adjusted the discussion accordingly.

16. Line 174: How can we see this claim in the presented data? It is not clear how it is supported. Figure 2 is so derived (using the difference) so it is very hard to see any such effects.

We now adjusted the figure to directly display the individual model estimates for the rate x motor effector interaction. Note, that due to the filtering changes suggested by Reviewer 2, now the random slopes for the rate are included, see R2 – Detailed issue 35.

17. Line 185: What is the Kaiser-Guttman criterion. Please offer a citation or explanation.

We explained the criterion and added the reference.

I. 573: “According to the Kaiser-Guttman criterion, all components that display eigen values exceeding 1 are selected^{99,100}.”

18. Line 188: Component labels are a bit tricky especially that the “Slow Whispering” PC which also loads on slow tapping to syllables and fast whispering to tones.

We agree that naming the components remains an approximation. We clarified this in the methods.

I. 576: “Based on the pattern of loadings (i.e., reflecting correlations) of the synchronization conditions on the rotated components, component labels were assigned. Component labels denote the synchronization conditions that showed the highest loading and therefore are simplifications of the complex dependencies.”

19. Table 3: How about the Slow Whispering PC?

The slow whispering component did not explain a significant share of variance in the step-wise regression procedure and it was therefore not included in the model. We clarified this in the results.

I. 244: “The slow whispering component did not explain a significant share of variance in the step-wise regression procedure and it was therefore not included in the model.”

20. Line 203: Sorry to nitpick, but there is no kind of 'optimality' shown here, just things that are relatively better or worse.

“Optimality” refers to the theoretic assumption, which was operationalized here with better performance. We clarified this in the introduction and phrased this more carefully throughout the manuscript:

L. 133: “The optimal rate was defined as the stimulus presentation rate with highest performance.”

21. Line 213: Please specify how “improved synchronization performance” and “increased auditory-motor synchronization performance” are defined. This is particularly important for making a conclusion about the correlates between synchronization and perception along the whole paper.

We rephrased the sentences as follows:

I. 239: “As expected, the GLMM additionally revealed significant fixed effects of the fast synchronization component and the slow tapping component indicating that perception performance was positively influenced by synchronization performance. That means that a better synchronization performance, as defined by a higher PLV, predicted higher auditory perception performance.”

22. Line 221: Please specify which are the “results described above”.

We deleted the sentence, since it referred to the envelope control analyses that were now moved to the main manuscript.

23. Line 235: What is the cut-off point between slow and fast rate? Maybe worth mentioning in the Introduction.

We used rates ranging from 1.92 to 2.08 Hz for the slow conditions and rates from 4.3 to 4.7 Hz in the fast conditions. We added this clarification to the introduction:

I. 160: “All tasks were performed at slow rates around 2 Hz (1.92 – 2.08 Hz) and fast rates around 4.5 Hz (4.3 – 4.7 Hz).”

24. Line 257: Can the authors really claim the activation of respective motor systems with the current design? Please clarify the reasoning behind.

The sentence has been revised and we now clarified our reasoning about the involvement of the motor system throughout the manuscript

I. 282: “A possible interpretation is that speech and music signals activate cortical rhythmic timing circuits with different optimal rates, resulting in better processing at these rates.”

25. Line 266: Ref 13 sounds cool, and is very relevant, but it is unpublished!

The nature journals state on their homepage that the citation of preprints is allowed. For transparency, we now mark all preprints as preprints.

<https://www.nature.com/nature-portfolio/editorial-policies/preprints-and-conference-proceedings>):

26. Line 268: Important reference to consider, on the surface seems like it might be contradictory: Zuk, N. J., Murphy, J. W., Reilly, R. B., & Lalor, E. C. (2021). Envelope reconstruction of speech and music highlights stronger tracking of speech at low frequencies. *PLoS computational biology*, 17(9), e1009358.

Thank you for pointing out the reference! We discuss the reference in our response to R1 - Issue 6 in detail. In summary, parts of the findings of Zuk et al. (2021) might be in line with our findings, while others are contradictory. The study (1) uses a different approach (e.g., much broader overlapping frequency ranges, passive listening, EEG etc.), and (2) the stimulus acoustics show no peak in the modulations around the beat rate (as typically shown in large corpus studies), and (3) the neural findings are in contrast to several other studies. Therefore, we now cite the study, however, refrain from detailed comparisons.

I. 330: “Furthermore, a common mechanism for the neural tracking of speech and music at faster rates has been suggested (with other findings of this study, however, being in contrast to ours and direct comparisons being hindered because of broader frequency ranges and other methodological differences)⁷⁵. “

I. 95: “Music contains rhythmic fluctuations beyond the beat rate at faster single note rates or slower phrasal rates^{18,75}.”

27. Line 277: Is this claim about domain-general mechanisms at slow time scale potentially contradictory the finding mentioned in Line 295? Please discuss more.

Our apologies for being not clear about this and thank you for pointing out the inconsistency.

Here (line 277 in previous version), we referred to the LMM conducted on the synchronization task data, where we observed a fixed effect of rate. Synchronization was generally better at slow rates than at fast rates, across motor effectors and stimuli. We had referred to this finding as indicating a domain-general mechanism that facilitates synchronization at slow rates. Additionally, the fixed effect was specified by an interaction of rate x motor effector, indicating distinct optimal synchronization for the different effector systems at the different rates (synchronization performance was only better for tapping compared to whispering at slow but not at fast rates). We now clarify in the manuscript that, from the fixed effect in the LMM, we cannot claim domain-general mechanisms at slow rates: Although we observe that on average participants are better in synchronizing at slow rates, the LMM does not reveal whether the synchronization for the different motor effectors reflects dependent mechanisms. Fig. 2 illustrates nicely that synchronizing at slow rates is easy for everybody, however, there are tight non-overlapping distributions for tapping and whispering at these rates. This is also indicated by our PCA, which allows to make claims about dependence/independence of components. The PCA indicates domain general (i.e., dependent) mechanisms at fast, but not at slow rates. This interpretation is further supported by our finding that the PCA component reflecting synchronization at fast rates independent of the speech/music domain highly correlated with musical sophistication. This is in line with a recent study showing that synchronization performance at 4.5 Hz was correlated for conditions with different motor effectors (Mares et al., 2023), and with previous research showing a correlation with musical sophistication (Assaneo et al. 2019; Rimmele et al. 2022).

We clarified this throughout the manuscript.

L. 320: “Although the results from the mixed effects model indicate that overall synchronization was better at slow rates, the PCA revealed no evidence that this reflects domain-general processes shared across motor effectors. Visual inspection of the mixed model predictions (Fig. 2) shows tight non-overlapping distributions for the synchronization of finger-tapping and whispering at slow rates. In contrast, the distributions were overlapping at fast rates. Accordingly, at fast rates, individuals with better whispering synchronization performance also showed better finger-tapping performance, resulting in one PCA component. This tentatively indicates that there exist domain-general influences that drive synchronization ability at fast rates.”

28. Line 283: Caution against using the terms “speech articulator” and the “music articulator” since first, the stimuli used in this study do not necessarily represent the music and speech; second, the articulators regarding these two domains are not limited to the articulators chosen by the authors in this paper.

We agree that it is a simplification. Our reasoning behind the choice for tapping and whispering is now in detail discussed in our responses to R2 – issue 2. We therefore specified the sentence as follows.

I. 300: “Although synchronization performance for motor effectors associated with speech (i.e., mouth and vocal cord) remains challenging at fast rates, whispering synchronization was less disadvantaged compared to finger-tapping at fast rates. “

29. Line 309-312: I may be dense but I don't quite see the argument here.

We deleted that section due to the thorough revision of the discussion. We removed the argument that our findings point towards the frequency of the brain rhythms involved, instead we just report proposals from the literature.

I. 51: “More specifically, the temporal processing of rhythmic structure in speech and music has been related to endogenous brain rhythms that show rhythmicity in the same frequency range as the speech and music signals ^{16,26-31}.”

I. 56: “Endogenous brain rhythms may support predictive processing and event segmentation by entraining to the rhythmic temporal modulations in the speech ^{20,33-35} and the music signal ^{18,36-38}.”

I. 284: “On the neural level, such optimal processing rates have been related to preferred auditory and motor cortex brain rhythms in the same frequency range ^{16,31}.”

30. Line 365: It would be great to keep the terms consistent e.g. perception vs. production task or perception vs. synchronization task

Thank you for pointing this out. We replaced production with synchronization in all occurrences.

31. Line 378: Curious to know why the authors didn't include the baseline condition such as hearing without performing and performing without hearing (this condition is important to understand the physical limitation of the articulators' movement, and assessing individual differences).

As the reviewer mentions, it is relevant to account for physical limitations. Therefore, we included the perception (hearing without performing) task in addition to the synchronization task. We did not include the baseline “performing without hearing”, i.e., a spontaneous tapping or whispering task. We now more thoroughly reviewed evidence on spontaneous rates in speaking, whispering, and tapping and their ranges (e.g., I.111 ff.; see our response to R2 – Issue 2 for details). We conclude from the available evidence that tapping and whispering spontaneously occurs around the respective optimal rates we selected. Additionally, tapping and whispering have been repeatedly shown to be possible at all rates used in our study. We agree that assessing the spontaneous rates would have provided an interesting addition to the study, but had decided against it due to the considerable length of the experimental session and the already available evidence.

32. Line 396: How did the authors synchronize the recorded whispers and taps to the auditory stimuli? Please add the details to the text.

We used an audiocard (RME Fireface UC) with high precision and presented stimuli using the full duplex mode implemented in the Psychophysics Toolbox. This mode supports simultaneous sound presentation and multi-channel audio capture. We recorded the presented stimulus with a loopback microphone, which enabled us to simultaneously record the stimulus and the participant's tapping and whispering. Our procedure allows for recoding the audio and the microphone output without any temporal jitter.

These details were added to the methods.

I. 446: "We used an audiocard (RME Fireface UC) with high precision and presented stimuli using the full duplex mode implemented in the Psychophysics Toolbox^{107,108}. This mode supports simultaneous sound presentation and multi-channel audio capture without any temporal jitter. We recorded the presented stimulus with a loopback microphone, which enabled us to simultaneously record the stimulus and the participant's tapping and whispering."

33. Line 409: Maybe justify why using "TEH", is it repeated in the syllables used for the main task?

The syllable "TEH" was used in the priming because participants were instructed to repeatedly whisper the syllable "TEH" in all speech production conditions of the synchronization task. In the original SSS test the syllable "TAH" is used (Assaneo et al., 2019; Lizcano-Cortés et al., 2022), however, in our lab we had previously used "TEH" and, as expected, this makes no difference as shown by our replications of the initial SSS-test results (Assaneo, Rimmele et al., 2021; Kern et al., 2021; Rimmele et al., 2022; Lubinus et al., 2023). We realize that this information was missing from the section and added it.

I. 443: "In the whispering conditions, participants were instructed to repeatedly whisper the syllable "TEH"."

34. Line 413: Totally missed from the above why the sequences are accelerating. Never would have expected that. Why is it done? Is it just to ensure people don't memorize some tempo?

With the procedure, including the use of accelerating sequences, we follow the procedure of the explicit version of the SSS test (Lizcano-Cortés et al. 2022). The SSS test aims at measuring spontaneous speech auditory-motor synchronization behavior, in order to estimate auditory-speech-motor cortex connection strength and avoid for example compensatory mechanisms that participants may use to boost their synchronization behavior (Assaneo et al., 2019). An initial version of the SSS test was implicit (i.e., no instruction to synchronize). Subsequently, an explicit version, that contains explicit instructions to synchronize has been suggested to similarly measure the spontaneous synchronization ability. Here, participants are unaware of the small undetectable rate acceleration.

We clarified this in the methods.

I. 467: "The synchronization sequences consisted of slightly accelerating tone or syllable sequences presented at fast or slow rates. This follows the established procedure of the explicit version of the SSS test⁹². Accelerating sequences are used to test for participants' spontaneous auditory-motor synchronization to slight, undetectable changes in the rate of the stimuli."

35. Line 450: Would have been much better for your arguments if your bandwidths had been properly scaled! The high bandwidth should have been 4.5 Hz wide not the same 2Hz as used at slow tempo.

We agree with the reviewer that applying the filtering in proportionally scaled bandwidths is more appropriate. As we closely want to match our methods to the SSS test, the bandwidth around the fast rates at 4.5 Hz is determined by the procedure used in the SSS Test that is performed at the same rate (Lizcano-Cortés et al. 2022). Therefore, instead of adjusting the bandwidth around the fast rates, we adjusted the bandwidth around the slow rates to apply a proportional scaling. Following the Reviewers' recommendations, we now use a filter from 1.56 to 2.44 Hz, which is the proportionally scaled to the filter range used at the fast rates (3.5 to 5.5 Hz). Adjusting the filter range resulted in some small changes, while the effects we reported were not altered: Likely due to a reduced noise level, the participant exclusion rates were slightly adjusted now resulting in a sample of 62 participants for the synchronization task (compared to 60 before the adjustment) and a sample of 57 participants for the perception task (compared to 56 before the adjustment). We observed some small changes in the results as a consequence of the adjustment. Most importantly, the synchronization LLM analysis now converged when adding a random slope for the rate, which was not possible before. We therefore added the random slope. As a consequence, the model explains a larger share of variance. The random slope effect is significant and the effects are displayed in a revised Fig. 2. All interactions and fixed effects previously included in the model remained significant. Additionally, the loadings in the PCA slightly changed, which did not lead to any changes in their interpretation or their effects on perception performance. We reported this in the methods and adjusted the results, tables, and figures accordingly.

L. 512: "For the fast sequences, filtering was applied between 3.5 and 5.5 Hz, following the procedure reported for the SSS test^{90,92}. For the slow sequences, the envelopes were filtered between 1.56 and 2.44 Hz."

36. Line 498: PCA on what? Please explain the procedure more completely.

We clarified this in the methods section:

I. 567: "The PCA aimed at summarizing the information from the individual PLVs in all eight synchronization conditions in a small number of principal components while retaining a sufficiently high share of the variance in synchronization performance. These components result from linear combinations of the observed variables (i.e., the PLVs of each participant in the eight synchronization conditions)."

37. Line 517: What was the recommendation? Why? It's overly opaque.

We added a clarification of the recommended model architecture.

I. 598: "The recommendations by Barr et al.¹¹² suggest that random slopes should be included on the subject level for within-subject predictors with several observations and their interactions. "

Decision letter and referee reports: second round

19th Oct 23

Dear Dr Barchet,

Thank you for your patience during the peer-review process. Your manuscript titled has now been seen by the same reviewers as before, and I include their comments at the end of this message. The reviewers find your work much improved with some minor concerns yet to be resolved. Editorially, there are also a few issues that need to be addressed before we can make a final decision on publication.

We therefore invite you to revise and resubmit your manuscript, along with a point-by-point response to the reviewers and editorial request. Please highlight all changes in the manuscript text file.

The primary editorial concern is that one of the key claims of the paper, i.e. that finger-tapping is advantaged compared to whispering at slow but not at faster rates is currently not sufficiently supported by the evidence. You present a significant interaction between effector and speed, and a non-significant main effect for differences between tapping and whispering at fast rates. However, the journal requires positive evidence for the absence of an effect to maintain this claim (and its discussion) in the manuscript. We therefore ask you to revise the work and provide Bayes factors or two-sided tests of equivalence to demonstrate that this key claim is supported by the evidence.

Reviewer #2 also highlights some concerns about the degree to which the results are appropriately interpreted and makes some valuable suggestions to this end, in particular in references to the passages in lines 200, 209, 378, 380. Please include a transparent limitations section (with its own sub-heading) as part of the Discussion section.

Please note that your revised manuscript must comply with our formatting and reporting requirements, which are summarized on the following checklist:

<https://www.nature.com/documents/commspsychol-style-formatting-checklist-article-rr.pdf> >{\$journal_name} formatting checklist and also in our style and formatting guide <https://www.nature.com/documents/commspsychol-style-formatting-guide-accept.pdf> >{\$journal_name} formatting guide .

To facilitate revisions and future processing, I also list a number of editorial requests at the end of this letter (after the Reviewer reports), which we ask you to address.

Please use the following link to submit your revised manuscript, point-by-point response to the referees' comments (which should be in a separate document to any cover letter) and the completed checklist:

<https://mts-commspsychol.nature.com/cgi-bin/main.plex?el=A3DU7BI7C3RI3I1A9ftdgUVaM4McxQEh9KGwOW5AZ>

Please do not hesitate to contact me if you have any questions or would like to discuss these

revisions further. We look forward to seeing the revised manuscript and thank you for the opportunity to review your work.

Best regards,

Dr Antonia Eisenkoeck
Senior Editor
Communications Psychology

EDITORIAL POLICIES AND FORMATTING

Editorial Policy: <https://www.nature.com/documents/nr-editorial-policy-checklist.pdf> Policy requirements (Download the link to your computer as a PDF.)

* **CODE AVAILABILITY:** All Communications Psychology manuscripts must include a section titled "Code Availability" at the end of the methods section. In the event of publication, we require that the custom analysis code supporting your conclusions is made available in a publicly accessible repository; at publication, we ask you to choose a repository that provides a DOI for the code; the link to the repository and the DOI will need to be included in the Code Availability statement. Publication as Supplementary Information will not suffice. We ask you to prepare code at this stage, to avoid delays later on in the process.

* **DATA AVAILABILITY:**

All Communications Psychology manuscripts must include a section titled "Data Availability" at the end of the Methods section or main text (if no Methods). More information on this policy, is available at <http://www.nature.com/authors/policies/data/data-availability-statements-data-citations.pdf>.

At a minimum the Data availability statement must explain how the data can be obtained and whether there are any restrictions on data sharing. Communications Psychology strongly endorses open sharing of data. If you do make your data openly available, please include in the statement:

We recommend submitting the data to discipline-specific, community-recognized repositories, where possible and a list of recommended repositories is provided at <http://www.nature.com/sdata/policies/repositories>.

If a community resource is unavailable, data can be submitted to generalist repositories such as

figshare or Dryad Digital Repository. Please provide a unique identifier for the data (for example a DOI or a permanent URL) in the data availability statement, if possible. If the repository does not provide identifiers, we encourage authors to supply the search terms that will return the data. For data that have been obtained from publicly available sources, please provide a URL and the specific data product name in the data availability statement. Data with a DOI should be further cited in the methods reference section.

Please refer to our data policies at http://www.nature.com/authors/policies/availability.html.

REVIEWERS' COMMENTS:

Reviewer #1 (Remarks to the Author):

Thank you for these revisions and interesting points of clarification. All of my questions have been addressed.

Benedikt Zoefel

Reviewer #2 (Remarks to the Author):

The authors have done a fine job at responding to our prior review. The paper is improved with relevant literature review, methodology details, additional analysis, clearly stated hypotheses and a clear focus on results. The additions about effectors and the role of acoustic differences were helpful. The analysis discussed on line 320 showing better performance at slow tempi yet still evidence for articulator differences and tying in the PCA was particularly clear.

One sentence in the abstract could use improvement as the 'and predicted by the synchronization performance' seems to oversimplify the results

"Perception of speech and music was better at different rates, and predicted by the synchronization performance."

Line 200: "Crucially, whispering synchronization was less disadvantaged compared with tapping at fast rates, which is consistent with research indicating that the speech production system is optimized at faster rates than the music production system" This is a stretch. This point is only driven by whispering synchronization being *_worse_* at slow rates, so calling it 'less disadvantaged' or 'optimized' for fast rates is arguably a distortion of the results, which clearly show sync is worse (on average) for whispering at fast rates than slow. The only observation I see that is literally consistent with your claim is that a small number of participants (6?) were better at fast rates when whispering only (positive slopes from slow to fast). This is a crucial argument for whispering to be optimal at fast rates, but I just don't see how it is supported by the data. Comes up again at Discussion Line 302.

Line 209: The PCA description is good, but I suggest you also note that PCs 2 and 3 have eigenvalues only marginally higher than one and thus do not capture much variance, so this reduces the strength with which you can make the desired claims. Suggested fix is to add a qualifier to the final summary sentence at the end of p 10. (Many ways to do so: a strong statement would be "although this conclusion is more tentative given the small eigenvalues for Components 2 and 3." Make sure this qualification is reflected in the discussion. For example Line 274: "with independent rhythmic timing mechanisms..." is stated too strongly, since it's largely based on small principal (not 'independent') components. Please moderate it.

Line 243 The use of motor PCs in the perceptual model is quite interesting, but I think could benefit from more motivation and discussion. Your conclusion "This emphasizes the importance of

motor contributions for temporal processing in the auditory domain" seems overstated. It is more appropriate, in my view to write "This suggests there is a link between motor and perceptual performance, is consistent with the importance of motor contributions..."

Line 247: excellent control of confounds!

Discussion

I think throughout the paper need to make a brighter line between behavioral results, and conclusions about neural mechanisms. Most of the discussion is well-qualified, but the occasional use of "This indicates..." (e.g. line 383) is too strong and I would prefer 'suggests' or 'is consistent with' or 'supports the idea'.

Line 304 "...the observed effects could result from peripheral constraints such as the ability to move our jaw faster than our fingers" – maybe I'm confused, but wouldn't that predict better synchronization at slow rates? Maybe revisit the wording, e.g. do you mean that the jaw may not be able to move slowly as well? Seems like the 'motor envelope' analysis could be quite relevant here.

Please make the final conclusion paragraph more specific to what was done:

Line 378. Saying 'speech perception was better' is too broad. 'perception of temporal deviants in speech' or even 'in a temporal deviant detection task, speech perception was better..' would better qualify it in my opinion.

Line 380 "synchronization of slow rates was independent for different motor effectors" – maybe I just missed it, but was there a specific test of independence (e.g. correlation analysis of finger vs whisper sync per individual) to directly show independence in a statistical sense? Please make this more precise in what sense you mean 'independent.'

R2 Issue-4 & 5. Sorry, I think my comment was misunderstood, due to my poor wording. It was clear you were not using polyphonic music; instead, I was talking more at the level of interpretation and cause about the possible origin for the dominant rates in speech and music. This is along the lines of some of Stephen Brown's work on the design features of speech and language. This preprint may also be of interest: [Redacted]

Detailed Editorial requests:

In your Reporting summary:

- 'For null hypothesis testing..' needs to be ticked and all relevant info needs to be added in the manuscript

- 'For Bayesian analysis..' needs to be ticked if you provide Bayes Factors rather than equivalence tests in response to the editorial request

- 'Estimates of effects sizes..' needs to be ticked and effect sizes need to be added in the manuscript

- sex/gender reporting - please specify whether participants self-reported gender or sex (both in the Reporting Summary and in the Manuscript). If they reported sex, then "female participants" and "male participants" is the preferred terminology; if they reported gender, we recommend use of "women", "men", "non-binary", "undisclosed gender", or similar as appropriate. If you instructed the participants in German (as data were collected in Germany), it would be beneficial to clarify what term you used and whether it is more likely to be understood in terms of (societal) gender or (biological) sex.

In addition, please undertake the following formatting revisions to your manuscript:

- Please delete the keywords

- Order of sections: Please follow the prescribed order of sections (see template), with the Methods following the Introduction and preceding the Results. Please include the heading "Introduction" and

a sub-heading in the Discussion "Limitations"

-Statistics need to be reported in full, including precise p values and effect sizes. We strongly recommend redundant reporting of key statistics in the tables and Results section.

-Please report whether data met assumptions of normality

-Please ensure that all figures plot data to the same scale (where measures are repeated across panels or figures) and that y-axes originate at or cut through 0.

-Please include the Data availability statement after the Methods

-Please prepare the code for sharing and reference to it in the Code availability statement

-Please include an author contributions statement

Response to Reviewers

REVIEWERS' COMMENTS:

Reviewer #1 (Remarks to the Author):

Thank you for these revisions and interesting points of clarification. All of my questions have been addressed.

Benedikt Zoefel

Thank you very much for the positive evaluation.

Reviewer #2 (Remarks to the Author):

The authors have done a fine job at responding to our prior review. The paper is improved with relevant literature review, methodology details, additional analysis, clearly stated hypotheses and a clear focus on results. The additions about effectors and the role of acoustic differences were helpful. The analysis discussed on line 320 showing better performance at slow tempi yet still evidence for articulator differences and tying in the PCA was particularly clear.

Thank you very much for the positive evaluation and the constructive feedback.

One sentence in the abstract could use improvement as the 'and predicted by the synchronization performance' seems to oversimplify the results

The sentence was revised.

I. 32:

"Perception of speech and music was better at different rates, and predicted by a fast general and a slow finger-tapping synchronization component."

Line 200: "Crucially, whispering synchronization was less disadvantaged compared with tapping at fast rates, which is consistent with research indicating that the speech production system is optimized at faster rates than the music production system" This is a stretch. This point is only driven by whispering synchronization being *worse* at slow rates, so calling it 'less disadvantaged' or 'optimized' for fast rates is arguably a distortion of the results, which clearly show sync is worse (on average) for whispering at fast rates than slow. The only observation I see that is literally consistent with your claim is that a small number of participants (6?) were better at fast rates when whispering only (positive slopes from slow to fast). This is a crucial argument for whispering to be optimal at fast rates, but I just don't see how it is supported by the data. Comes up again at Discussion Line 302.

We rephrased the sentence to better reflect our findings:

I. 454:

"Crucially, tapping synchronization was only advantaged compared with whispering at slow rates, which is consistent with research indicating that the music production system is optimized at slower rates than the speech production system¹³."

I. 563:

“Although synchronization performance for motor effectors associated with speech (i.e., mouth and vocal cord) remains challenging at fast rates, finger-tapping synchronization showed no advantaged compared to whispering at fast rates.”

Line 209: The PCA description is good, but I suggest you also note that PCs 2 and 3 have eigenvalues only marginally higher than one and thus do not capture much variance, so this reduces the strength with which you can make the desired claims. Suggested fix is to add a qualifier to the final summary sentence at the end of p 10. (Many ways to do so: a strong statement would be “although this conclusion is more tentative given the small eigenvalues for Components 2 and 3.” Make sure this qualification is reflected in the discussion. For example Line 274: “with independent rhythmic timing mechanisms...” is stated too strongly, since it’s largely based on small principal (not ‘independent’) components. Please moderate it.

As suggested by the reviewer, we included a moderating sentence at the end of the results paragraph:

I. 473:

“It should however be noted that this conclusion is tentative given the small eigen values for components 2 and 3.”

Line 243 The use of motor PCs in the perceptual model is quite interesting, but I think could benefit from more motivation and discussion. Your conclusion “This emphasizes the importance of motor contributions for temporal processing in the auditory domain” seems overstated. It is more appropriate, in my view to write “This suggests there is a link between motor and perceptual performance, is consistent with the importance of motor contributions...”

We revised the conclusion as suggested by the reviewer.

I. 505:

“This suggests a link between motor and perceptual performance and is consistent with previous work emphasizing the importance of motor contributions to perceptual performance in the auditory domain.”

Line 247: excellent control of confounds!

Thank you!

Discussion

I think throughout the paper need to make a brighter line between behavioral results, and conclusions about neural mechanisms. Most of the discussion is well-qualified, but the occasional use of “This indicates...” (e.g. line 383) is too strong and I would prefer ‘suggests’ or ‘is consistent with’ or ‘supports the idea’.

Following the reviewer’s suggestion, we replaced “indicates” with “suggests” in all occurrences, where an interpretation followed. Examples are I. 536, I. 554, and line 650.

Line 304 “...the observed effects could result from peripheral constraints such as the ability to move our jaw faster than our fingers” – maybe I’m confused, but wouldn’t that predict better synchronization at slow rates? Maybe revisit the wording, e.g. do you mean that the jaw may not be able to move slowly as well? Seems like the ‘motor envelope’ analysis could be quite relevant here.

We clarified the sentence.

L. 566:

“Alternatively, the observed effects could result from peripheral constraints for fast finger movements. The advantage of finger-tapping compared to whispering might be only present at slow but not at fast rates because of constraints that reduce the accuracy of synchronized finger-tapping at fast rates. However, peripheral constraints cannot account for our findings in the perception task in which no overt production was required.”

Please make the final conclusion paragraph more specific to what was done: Line 378. Saying ‘speech perception was better’ is too broad. ‘perception of temporal deviants in speech’ or even ‘in a temporal deviant detection task, speech perception was better...’ would better qualify it in my opinion.

We rephrased the sentence in order to describe the study’s conclusion more specifically:

I. 644:

“In conclusion, we show that discrimination of temporal deviants versus regular occurrences at faster rates was better in syllable sequences compared to tone sequences and the opposite was the case for slower rates.”

Line 380 “synchronization of slow rates was independent for different motor effectors” – maybe I just missed it, but was there a specific test of independence (e.g. correlation analysis of finger vs whisper sync per individual) to directly show independence in a statistical sense? Please make this more precise in what sense you mean ‘independent.’

We added a clarification in the methods section that our data met the assumption of multivariate normal distribution. As a result, the principal components can be regarded as independent.

I. 350

“The data met assumptions of multivariate normality based on a Henze-Zirkler test for multivariate normality (HZ = 0.95, $p = .353$). This implies that the extracted components can be regarded as uncorrelated and independent¹⁰⁷.”

Additionally, we revised the sentence in the conclusion to clarify that it refers to the principal component analysis.

I. 646:

“Our analysis of auditory-motor synchronization revealed that although performance was overall higher at slow rates, synchronization at slow rates was related to independent principal components for different motor effectors associated with speech and music.”

R2 Issue-4 & 5. Sorry, I think my comment was misunderstood, due to my poor wording. It was clear you were not using polyphonic music; instead, I was talking more at the level of interpretation and cause about the possible origin for the dominant rates in speech and music. This is along the lines of some of Stephen Brown’s work on the design features of speech and language. This preprint may also be of interest: <https://osf.io/preprints/psyarxiv/2635u>

Thank you for the clarification and the link to the references. We now added a reference to a study by Stephen Brown and colleagues (Savage et al., 2015) when we discuss the putative role of beat in ensemble coordination (I. 82). Thank you for pointing out the additional reference. We however refrain from citing it, as it goes beyond the scope of our discussion.

Other changes:

We discovered that, in our previous revision, we failed to completely revise the codes used for the envelope filtering. We corrected this error. As a consequence, our results remained the same, that is: all significances and included variables in the linear and logistic mixed models remained the same, the number of PCA components remained the same and the loadings only changed to a limited degree. We updated the tables and the figures accordingly and changed all numbers that were affected by the mistake. The changes are highlighted in the manuscript. We apologize for not realizing this mistake earlier.

Decision letter and referee reports: Third round

6th Dec 23

Dear Ms Barchet,

We have now had a chance to assess the revised version of your manuscript titled "Auditory-Motor Synchronization and Perception Suggest Partially Distinct Time Scales in Speech and Music", and I am delighted to say that we are happy, in principle, to publish a suitably revised version in Communications Psychology under the open access CC BY license (Creative Commons Attribution v4.0 International License).

We therefore invite you to revise your paper to comply with our format requirements and to maximise the accessibility and therefore the impact of your work.

EDITORIAL REQUESTS:

SUBMISSION INFORMATION:

OPEN ACCESS:

Communications Psychology is a fully open access journal. Articles are made freely accessible on publication under a [CC BY license](http://creativecommons.org/licenses/by/4.0) (Creative Commons Attribution 4.0 International License). This license allows maximum dissemination and re-use of open access materials and is preferred by many research funding bodies.

For further information about article processing charges, open access funding, and advice and support from Nature Research, please visit <https://www.nature.com/commspsychol/article-processing-charges>

At acceptance, you will be provided with instructions for completing this CC BY license on behalf of all authors. This grants us the necessary permissions to publish your paper. Additionally, you will be asked to declare that all required third party permissions have been obtained, and to provide billing information in order to pay the article-processing charge (APC).

* **CODE AVAILABILITY:** All Communications Psychology manuscripts must include a section titled "Code Availability" at the end of the methods section. We require that the custom analysis code supporting your conclusions is made available in a publicly accessible repository at this stage;

please choose a repository that generates a digital object identifier (DOI) for the code; the link to the repository and the DOI must be included in the Code Availability statement. Publication as Supplementary Information will not suffice.

* DATA AVAILABILITY:

<https://mts-commpsychol.nature.com/cgi-bin/main.plex?el=A1DU6BI7D5RI4I5A9ftdgUVaM4McxQEh9KGwOW5AZ>

Best regards,

Antonia Eisenkoeck

Antonia Eisenkoeck
Senior Editor
Communications Psychology